# Atomic-scale regulation of anionic and cationic migration in alkali metal batteries

Pan Xiong[1,2,6], Fan Zhang[2,6], Xiuyun Zhang[3,6], Yifan Liu[1], Yunyan Wu[1], Shijian Wang[2], Javad Safaei[2], Bing Sun [2], Renzhi Ma[4], Zongwen Liu [5], Yoshio Bando [4], Takayoshi Sasaki [4], Xin Wang[1], Junwu Zhu [1✉] & Guoxiu Wang [2✉]

The regulation of anions and cations at the atomic scale is of great significance in membrane-based separation technologies. Ionic transport regulation techniques could also play a crucial role in developing high-performance alkali metal batteries such as alkali metal-sulfur and alkali metal-selenium batteries, which suffer from the non-uniform transport of alkali metal ions (e.g., $Li^+$ or $Na^+$) and detrimental shuttling effect of polysulfide/polyselenide anions. These drawbacks could cause unfavourable growth of alkali metal depositions at the metal electrode and irreversible consumption of cathode active materials, leading to capacity decay and short cycling life. Herein, we propose the use of a polypropylene separator coated with negatively charged $Ti_{0.87}O_2$ nanosheets with Ti atomic vacancies to tackle these issues. In particular, we demonstrate that the electrostatic interactions between the negatively charged $Ti_{0.87}O_2$ nanosheets and polysulfide/polyselenide anions reduce the shuttling effect. Moreover, the $Ti_{0.87}O_2$-coated separator regulates the migration of alkali ions ensuring a homogeneous ion flux and the Ti vacancies, acting as sub-nanometric pores, promote fast alkali-ion diffusion.

[1] Key Laboratory for Soft Chemistry and Functional Materials of Ministry Education, Nanjing University of Science and Technology, Nanjing, China. [2] Centre for Clean Energy Technology, School of Mathematical and Physical Sciences, Faculty of Science, University of Technology Sydney, Broadway, Sydney NSW 2007, Australia. [3] College of Physical Science and Technology, Yangzhou University, Yangzhou, China. [4] International Center for Materials Nanoarchitectonics (WPI-MANA), National Institute for Materials Science (NIMS), Tsukuba, Ibaraki, Japan. [5] School of Chemical and Biomolecular Engineering, The University of Sydney, Sydney, NSW, Australia. [6]These authors contributed equally: Pan Xiong, Fan Zhang, Xiuyun Zhang. ✉email: zhujw@njust.edu.cn; guoxiu.wang@uts.edu.au

Rechargeable batteries beyond lithium-ion chemistry are considered as the most promising candidates for next-generation energy storage with low cost and high energy densities[1–3]. Among them, alkali metal batteries such as alkali metal-sulfur and alkali metal-selenium batteries have attracted much attention due to their high theoretical energy densities[4–7]. However, multiple obstacles associated with both S and Se cathodes and alkali metal anodes have severely hindered their practical applications, especially the unwanted shuttle effect of soluble polysulfide/polyselenide (PS) intermediates and the formation of alkali metal dendrites, which originate from the uneven migration of PS anions and alkali metal cations[8–10]. Considerable efforts have been devoted to mitigating these detrimental effects, including the use of composite S/Se cathodes[7,11–14], modified metal anodes[15,16], functionalized separators[17–26], and solid electrolytes[27–29]. Although these efforts have achieved some success by either impeding the shuttling effect or suppressing dendrite growth, the performance of lab-scale cells obtained so far is still far from satisfactory. Therefore, it is desirable to develop a multifunctional approach which can simultaneously regulate both cationic and anionic migrations for alkali metal batteries.

In a battery system, separator membranes provide channels for diffusion of both cations and anions and also act as an electronic insulating material to prevent cell short circuit. In alkali metal-S/Se batteries, nano- and micro-porous polyolefin-based membranes such as polypropylene (PP), are commonly used as separators. However, these porous membranes with a large pore size have a limited selective effect on the transportation of ions[30]. The flow of both alkali metal ions (such as $Li^+$ and $Na^+$) and PS anions can be considerably throttled when using such separator membranes. On the one hand, small but cumulative diffusion of PSs usually causes irreversible loss of cathode active materials, resulting in poor cycle life of batteries. Furthermore, unregulated diffusion of alkali metal ions induces inhomogeneous alkali metal deposition, leading to possible formation of metal dendrites and short-circuiting. Hence modification of separators could be a straightforward way to control the migration of both alkali metal cations and PS anions to simultaneously eliminate the formation of metal dendrites and the shuttle effect of PSs. So far, various functional materials have been employed to modify separators, including carbon materials[18,21,31–36], metal-based oxides[22,24,37,38], sulfides[39–41], carbides[42] and hydroxides[43,44], and metal-organic frameworks (MOFs)[17,20,45–49]. However, to effectively suppress PS shuttling, most of the functional layers used to date have a high weight density and a large thickness. These inevitably place a severe burden on the weight and useable volume of the whole cell, which subtracts from the targeted high energy densities of alkali metal-S/Se batteries. More importantly, the retarding layer is an extra barrier to ion transfer, which causes large interfacial resistance and suppresses the transport of alkali metal cations. Therefore, an ideal functional layer should be as thin as possible to maximize alkali metal cation transport without compromising the ability to prevent PS shuttling, thus forming a selective ionic sieve with high permeability for alkali metal ions.

Permselective ionic sieves have been widely applied in membrane-based separation technologies mainly based on the basic size-sieving effect and electrical interaction between ions and the membrane[50–52] (Fig. 1a). Ions/molecules with sizes smaller than the pore size of the membrane can pass, while the ions with larger volumes are selectively excluded. When significant electrical interactions are present, charged membrane surfaces repel identically charged ions while attracting oppositely charged ions which may then permeate through the membrane. Because the flux of ions is inversely proportional to the membrane thickness, an 'ultimate' membrane would be a one-atom-thick layer with

well-defined nanopores[53]. Therefore, two-dimensional (2D) porous materials with an atomic-scale thickness have attracted extensive interest[54–56]. Since 2D atomically thin nanosheets with sub-nanometer pores could act as a highly selective and permeable separator, they are highly desirable for long-life alkali metal-S/Se batteries. Nanometric materials with structural bidimensionality had been investigated, including graphene oxide (GO)/reduced graphene oxide (rGO), $MoS_2$ and MXenes. The membranes/separators of this type that have been studied were generally thick layers ranging from several micrometers to hundreds of micrometers[32,39,42]. The PS shuttling effects were mitigated owing to the steric hindrance effect, but the diffusion of Li or Na-ions was also hindered. Furthermore, without nanopores (1–100 nm), the $Li^+/Na^+$ ions could only migrate through gaps between layers of the materials. Therefore, fabrication of 2D atomically thin membranes with sub-nanometer pores (<1 nm) is a niche strategy for attaining higher performance alkali metal batteries (Fig. 1b). To the best of our knowledge, this has not yet been reported. Sub-nanopores could efficiently exclude large PS anions while allowing small $Li^+/Na^+$ ions to rapidly pass through. A nanometric membrane would minimize main transit lengths. Moreover, using negatively charged nanosheets can repel anionic PS ions and thus enhance the suppression of the shuttle effect while simultaneously facilitating the diffusion of $Li^+/Na^+$ cations via the membrane-ion electrical interactions.

Here, we report 2D negatively charged titania ($Ti_{0.87}O_2$) nanosheets with Ti atomic vacancies as a selective ionic sieve in alkali metal batteries. The $Ti_{0.87}O_2$ nanosheets, delaminated from their parent layered oxides, are single-crystal unilamellar nanosheets with a thickness of 0.75 nm. By a facile filtration method, these unilamellar layers were layer-by-layer self-assembled onto commercial PP separators to form a $Ti_{0.87}O_2$ coating layer with a controlled thickness of ~80 nm and a surface area mass loading of 0.016 mg cm$^{-2}$. The Ti atomic vacancies can act as sub-nanometer pores, making the $Ti_{0.87}O_2$ layer a promising separator coating material for simultaneously regulating the migration of $Li^+/Na^+$ cations and PS anions at an atomic scale. At the anode side of alkali metal batteries (Fig. 1c), the negatively charged nanosheets offer strong electrostatic interaction for the efficient adhesion and homogeneous distribution of $Li^+/Na^+$ ion flux, resulting in reducing the growth of Li/Na metal deposition with adverse morphologies. Moreover, the rich Ti vacancies and nanometric thickness provide fast pathways for the diffusion of $Li^+/Na^+$ cations. At the S/Se cathode side of alkali metal batteries (Fig. 1d), the negatively charged $Ti_{0.87}O_2$ nanosheets with a high negative charge density effectively exclude PS anions via a strong electrostatic repulsion effect. Besides, the PS anions with sizes larger than the size of the Ti vacancies are selectively excluded because of the geometrical restrictions. As a result, when applied in Li-S, Li-Se, and Na-Se batteries, the $Ti_{0.87}O_2$-coated separators enable long-term cycling stability. Flexible single-layer Li-S pouch cells (6.0 cm × 6.5 cm) were fabricated and exhibited stable cycling performance under different bending conditions, demonstrating the potential of the $Ti_{0.87}O_2$ nanosheets for practical applications.

## Results and discussion

### Fabrication and characterizations of the functional $Ti_{0.87}O_2$/PP separator.
Negatively charged $Ti_{0.87}O_2$ nanosheets in the form of $Ti_{0.87}\square_{0.13}O_2$, where $\square$ represents the Ti vacancies, were prepared by soft chemical exfoliation of layered lepidocrocite-type titanate crystals[57,58]. As illustrated in Fig. 2a, the nanosheet is a single-crystal-like 2D ultrathin monolayer (0.75 nm thickness) with a high density of Ti vacancies[59]. The Ti atomic vacancy is the octahedral site, which can accommodate a $Li^+$ ion (0.76 Å in radius) or $Na^+$ ion (1.02 Å in radius) but smaller than a PS

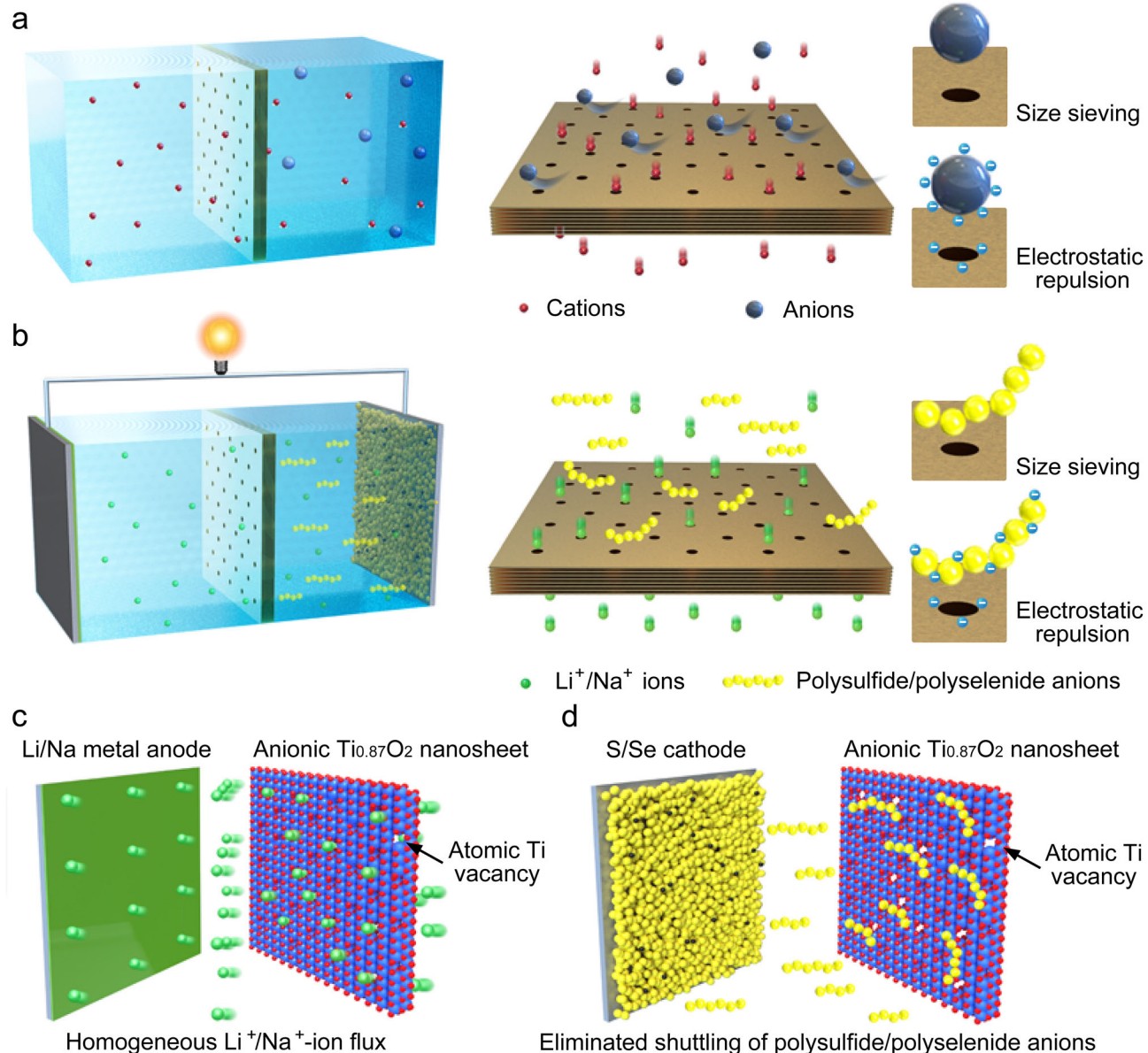

**Fig. 1 Schematic illustration of 2D porous nanosheets as a selective ionic sieve in membrane-based separation and alkali metal-S/Se batteries.**
**a** 2D porous nanosheet-based membranes act as selective ionic sieves in membrane-based separation technologies. The small cations can pass, while large anions are selectively excluded based on a size-sieving effect and electrical interaction between ions and membrane. **b** 2D nanosheets with sub-nanometer pores act as selective ionic sieves in alkali metal-S/Se batteries. The small $Li^+/Na^+$ ions can rapidly pass through the sub-nanopores, while the large polysulfide/polyselenide (PS) anions are selectively excluded due to size-sieving effect and electrical repulsion between PS anions and negatively charged nanosheets. **c** 2D negatively charged $Ti_{0.87}O_2$ nanosheets with atomic Ti vacancies offer strong electrostatic interaction with $Li^+/Na^+$ ions, resulting in homogeneous distribution of $Li^+/Na^+$-ion flux, preventing the growth of Li/Na dendrites. **d** 2D negatively charged $Ti_{0.87}O_2$ nanosheets with a high negative charge density provide a strong electrostatic repulsion of PS anions, resulting in effective suppression of PS shuttling.

anion[60,61]. Therefore, the Ti vacancies may work as migration-aids for $Li^+/Na^+$ ions and obstacle channels for PS anions. The diffusion of $Li^+$ ions through the $Ti_{0.87}O_2$ nanosheets deposited on a cathode material has been reported in our previous report[62]. Atomic force microscopy (AFM) analysis (Fig. 2b) confirmed that the exfoliated $Ti_{0.87}O_2$ nanosheets are unilamellar sheets with a uniform thickness of ~1.1 nm. Transmission electron microscopy (TEM) images as shown in Fig. 2c display a flat and transparent sheet-like morphology, which is consistent with the AFM observation. Selected area electron diffraction (SAED) (inset in Fig. 2c) indicates the mono-crystalline nature of the $Ti_{0.87}O_2$ nanosheets. Figure 2d shows an atomic-resolution high-angle annular dark-field scanning transmission electron microscopy

(HAADF-STEM) image of a $Ti_{0.87}O_2$ nanosheet, where the Ti vacancies can be clearly visualized[61,63]. The Ti vacancies endow the obtained nanosheets negative charges, which has been confirmed by zeta-potential measurements (Supplementary Fig. 1). X-ray absorption fine spectroscopy (XAFS) was conducted to further investigate the structural characteristics of the defect-containing nanosheets. Figure 2e shows the Ti K-edge X-ray absorption near-edge structure (XANES) spectra of commercial rutile $TiO_2$ and $Ti_{0.87}O_2$ nanosheets. The pre-edge peak at ~4981 eV represents transitions of core electrons into O 2p states that are hybridized with the empty Ti 4p state[64,65]. The intensity of this peak for the $Ti_{0.87}O_2$ nanosheets was increased compared to that of rutile $TiO_2$, elucidating a decreased electron number of the

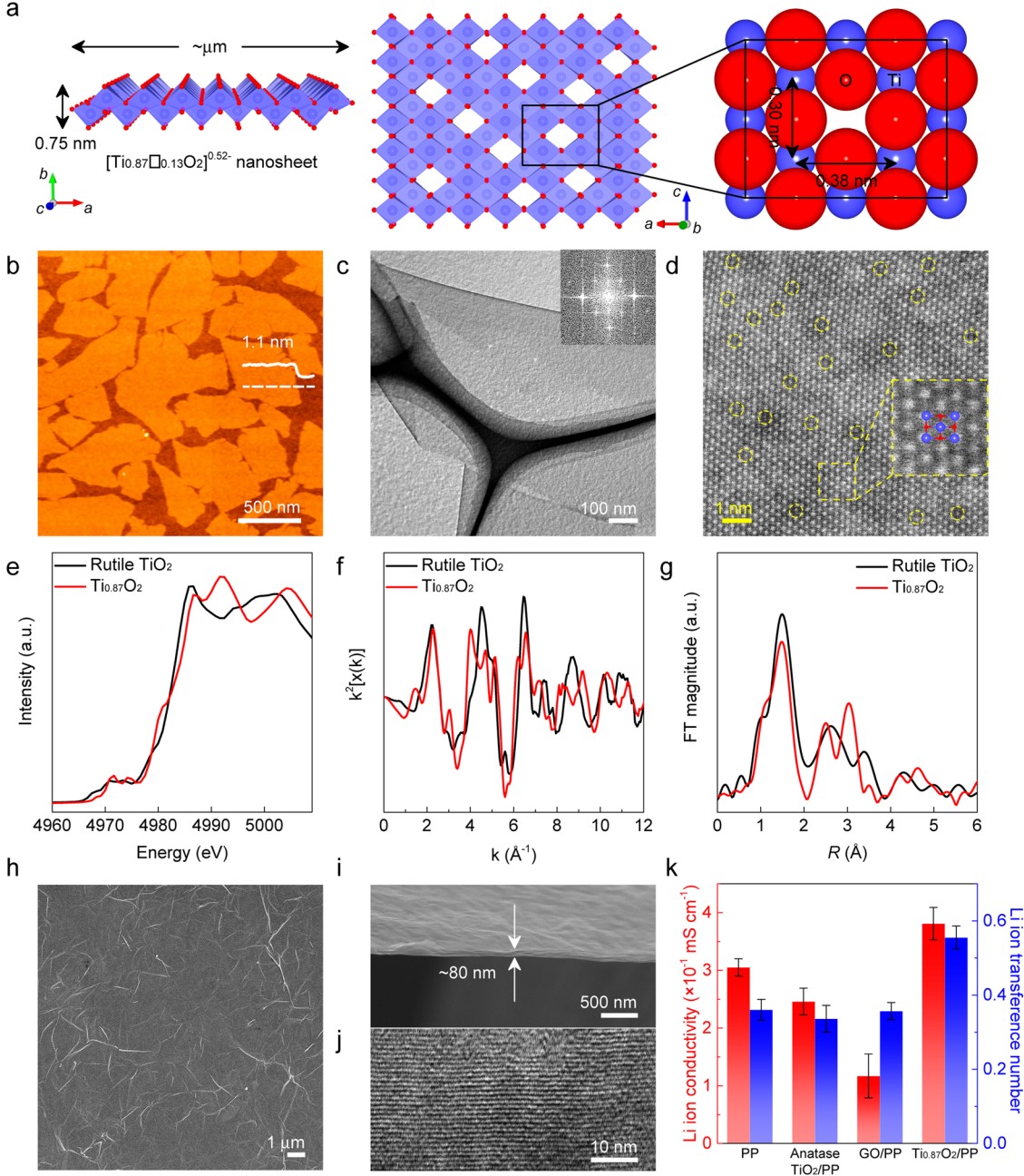

**Fig. 2 Fabrication and characterization of Ti$_{0.87}$O$_2$ nanosheets and Ti$_{0.87}$O$_2$/PP separators. a** Crystal structures of Ti$_{0.87}$O$_2$ nanosheet with respect to $c$- and $b$-axes. Enlarged structure using ionic radii to show the relative size of the Ti vacancy. **b** AFM image of Ti$_{0.87}$O$_2$ nanosheets. The nanosheets were deposited onto Si wafer substrates. **c** TEM image and SAED pattern of a Ti$_{0.87}$O$_2$ nanosheet. **d** HAADF-STEM image of a Ti$_{0.87}$O$_2$ nanosheet. **e** X-ray absorption near edge structure (XANES) of Ti K edge for commercial rutile TiO$_2$ and a freeze-dried sample of Ti$_{0.87}$O$_2$ nanosheets. **f** The $k^3$-weighted EXAFS in $K$-space for commercial rutile TiO$_2$ and a freeze-dried sample of Ti$_{0.87}$O$_2$ nanosheets. **g** Fourier transforms of $k$-space oscillations of Ti K edge of commercial rutile TiO$_2$ and a freeze-dried sample of Ti$_{0.87}$O$_2$ nanosheets. **h** SEM image showing a top-down view of a Ti$_{0.87}$O$_2$/PP separator. **i** SEM image showing a side-on view of a Ti$_{0.87}$O$_2$/PP separator. **j** TEM image showing a side view of a Ti$_{0.87}$O$_2$/PP separator. **k** Li-ion conductivity and Li-ion transference number of PP, anatase TiO$_2$/PP, GO/PP, and Ti$_{0.87}$O$_2$/PP separators. Error bars were included, which represent the standard deviation of the data taken from five samples.

O 2p–Ti 4p hybrid orbitals. This result indicates the presence of lattice O atoms with unsaturated coordination, which should be attributed to the presence of nearby Ti vacancies[66]. Moreover, the Ti K-edge extended XAFS (EXAFS) $k^3$x($k$) oscillation curve of the Ti$_{0.87}$O$_2$ nanosheets exhibited a slight intensity decrease compared to rutile TiO$_2$ (Fig. 2f), which also confirms the presence of Ti vacancies[65,66]. The interatomic distances of rutile TiO$_2$ and Ti$_{0.87}$O$_2$ nanosheets were determined through Fourier

transformed Ti K-edge EXAFS data (Fig. 2g). The first major coordination peak corresponds to the nearest Ti-O bond in the first coordination shell. The peak intensity for the Ti$_{0.87}$O$_2$ nanosheets obviously decreased relative to the TiO$_2$ samples, which further verifies the presence of Ti vacancies[65,66].

Functional Ti$_{0.87}$O$_2$/PP separators were fabricated by a facile vacuum filtration method (Supplementary Fig. 2). Compared to the porous surface of PP separators (Supplementary Fig. 3),

$Ti_{0.87}O_2$/PP separators showed a homogeneous morphology with uniform surface coverage by $Ti_{0.87}O_2$ nanosheets (Fig. 2h). Curved nanowrinkles without cracks were observed for the obtained $Ti_{0.87}O_2$ layers on PP separators as shown in the SEM image of Fig. 2h, which often results in additional and wider transporting channels for increased permeability[67,68]. As shown in Supplementary Fig. 4, a broad 010 diffraction peak of lamellar $Ti_{0.87}O_2$ was observed, which is similar to that from the nanosheets being self-assembled onto commercial PP separators. An XRD analysis of $Ti_{0.87}O_2$ nanosheets without PP separator was also conducted. As shown in Supplementary Fig. 5, two diffraction peaks appeared in a low angular range can be indexed as 010 and 020. This indicates a lamellar structure with a gallery height of ~1.1 nm, which is consistent with the results of self-assembled $Ti_{0.87}O_2$ nanosheets on PP separators. Due to the negative charge on the surface, some positively charged species should be included in the $Ti_{0.87}O_2$ nanosheets to make the whole charge balance, such as proton and tetrabutylammonium ($TBA^+$) ions, which were used in the synthetic process of the $Ti_{0.87}O_2$ nanosheets[69]. Almost all $TBA^+$ ions initially trapped between the nanosheets could be decomposed upon exposure to UV light. In order to analyze the composition, the weight loss of the final nanosheet films without PP separators was recorded during a heating process in air. As shown in Supplementary Fig. 6, the weight loss before 250 °C is associated with the liberation of structural $H_2O$ molecules; while the decomposition of interlayer $TBA^+$ ions could result in the mass loss between 250 and 500 °C. Then, after a rough calculation, the final composition of the nanosheet films in our work was qualitatively determined to be $H_{0.98}TBA_{0.09}Ti_{1.73}O_4 \cdot 1.58H_2O$. Considering that the interlayer distance of 1.1 nm is too small to accommodate $TBA^+$ ions, the small amount of $TBA^+$ ions should be included in a gap between the restacked $Ti_{0.87}O_2$ layers.

The surface area mass loading and thickness of the $Ti_{0.87}O_2$ functional layers in the resultant $Ti_{0.87}O_2$/PP separator can be conveniently controlled by directly adjusting the volume of the nanosheet suspensions used in the vacuum filtration process. Supplementary Fig. 7 shows X-ray diffraction (XRD) data of $Ti_{0.87}O_2$/PP separators with different surface area mass loadings. As the weight density increased, the XRD reflections became more intense, with increasing thickness of the $Ti_{0.87}O_2$ layer. A cross-sectional scanning electron microscopy (SEM) image (Fig. 2i) shows a coating thickness of ~80 nm. The surface area mass loading of the $Ti_{0.87}O_2$ nanosheets in the $Ti_{0.87}O_2$/PP separator was estimated to be ~0.016 mg cm$^{-2}$. Cross-sectional TEM images displayed parallel lamellar fringes (Fig. 2j), further revealing the layer-by-layer assembly of the $Ti_{0.87}O_2$ nanosheets. The fringe spacing was measured to be ~1.1 nm, which is consistent with the basal spacing in the XRD pattern (Supplementary Fig. 4). It should be noted that the weight and thickness of the $Ti_{0.87}O_2$ layer was only ~0.32% and 1.5% of those of the commercial PP separator (thickness, 25 μm; weight, 2.16 mg; diameter, 16 mm), respectively. Such a low surface area mass loading and ultrathin thickness have not been reported previously, to the best of our knowledge (Supplementary Table 1). The as-prepared nanometric coating $Ti_{0.87}O_2$ layers can permit significantly high cation fluxes, which results in fast Li/Na-ion diffusion. The morphology and cross-sectional characteristics of $Ti_{0.87}O_2$/PP separators with relatively high surface area mass loading of 0.032 and 0.096 mg cm$^{-2}$ were also investigated (Supplementary Figs. 8–11). Homogenously stacked layers with thicknesses of ~150 and 460 nm were obtained, respectively.

For comparison, anatase $TiO_2$/PP and GO/PP separators with the same surface area mass loading of 0.016 mg cm$^{-2}$ were fabricated (Supplementary Figs. 2 and 4). As shown in Supplementary Fig. 12, the anatase $TiO_2$ nanoparticles did not disperse uniformly when coated onto PP surfaces. Only a limited part of the PP surfaces was covered by the aggregates of anatase $TiO_2$ nanoparticles. This is clearly different from the charged nanosheets in suspensions where aggregation has been prevented due to Coulombic repulsion (neutral nanoparticles with high surface energy are prone to aggregate). As another negatively charged nanosheet material, GO was able to uniformly coat on the surface of PP separators (Supplementary Fig. 13). A slightly larger thickness of ~120 nm was observed for the GO/PP separator (Supplementary Fig. 14), compared with the $Ti_{0.87}O_2$/PP separators. This matches the calculated result based on an ideal 2D theoretical specific surface area in the lateral dimensions of GO and $Ti_{0.87}O_2$ monolayers (Supplementary Fig. 15).

The coating of nanometric $Ti_{0.87}O_2$ layer brings several advantages to improve the electrochemical performance of a separator membrane. As shown in Supplementary Fig. 16, after being placed on a hot plate and heated at 120 °C for 10 min in air environment, the $Ti_{0.87}O_2$/PP film retained its original geometrical shape, while the pristine PP film tended to shrink. The improved thermal stability of the $Ti_{0.87}O_2$/PP would benefit the safety of batteries in practical applications. Supplementary Fig. 17 shows the contact angles of electrolyte (1 M LiTFSI in DME: DOL 1: 1, v/v) on the PP and $Ti_{0.87}O_2$/PP separators. A smaller contact angle was observed on the $Ti_{0.87}O_2$/PP separators than that on PP separators, suggesting a better wettability of the $Ti_{0.87}O_2$/PP separators by electrolyte. Although the nanosheets covered the wide pores of PP separators, the improved wettability is beneficial for accelerating the electrolyte penetration, thus facilitating the transport of $Li^+$ ions[21]. Besides this effect, the as-prepared $Ti_{0.87}O_2$/PP separators showed high stability under various degrees of mechanical bending (Supplementary Fig. 18). This suggested a strong adhesion between $Ti_{0.87}O_2$ nanosheets and PP separators, which may be ascribed to the van der Waals interaction produced by the poly(tetrafluoroethylene) (PTFE) binder (PTFE was used during the slurry preparation of the separator coating mixture). As shown in Supplementary Figs. 19–21 and Fig. 2k, the $Li^+$ ion conductivity (see the Experimental details in the Supplementary Information) of the $Ti_{0.87}O_2$/PP separators (0.381 ± 0.028 mS cm$^{-1}$) was higher than that of the bare PP (0.305 ± 0.015 mS cm$^{-1}$) and anatase $TiO_2$/PP separators (0.246 ± 0.023 mS cm$^{-1}$), and over three times higher than GO/PP separators (0.117 ± 0.038 mS cm$^{-1}$). The Li-ion transference numbers were also determined, as shown in Supplementary Fig. 22 (see the Experimental details in the Supplementary Information). Compared to other functional layers, the $Li^+$ ion transference number increased significantly from 0.36 ± 0.03 for bare PP to 0.55 ± 0.03 for $Ti_{0.87}O_2$/PP with a surface area mass loading of 0.016 mg cm$^{-2}$ (Fig. 2k and Supplementary Fig. 23). As shown in Supplementary Figs. 21 and 23, the Li-ion conductivity and Li-ion transference number of the $Ti_{0.87}O2$/PP separator decreased when the surface area mass loading and thickness increased. An optimized $Ti_{0.87}O_2$/PP separator with a surface area mass loading of 0.016 mg cm$^{-2}$ was achieved with the largest Li-ion conductivity and Li-ion transference number. Supplementary Table 2 summarized some previously reported work on modified separators for Li-S batteries. Generally, covering open pores of pristine separators will increase the path of ion movement, leading to reduced Li-ion diffusion. Thus, as shown in the Supplementary Table 2, modification of pristine separators sometimes results in decrease of the $Li^+$ conductivity. However, the above testing results demonstrated that $Ti_{0.87}O_2$ nanosheets can facilitate Li-ion migration. Similar phenomenon of slightly increased conductivity for modified separators have also been observed in recent research works as shown in the Supplementary Table 2. Because the $Ti_{0.87}O_2$ layers are negatively charged with cation vacancies, the electrostatic attraction force

between $Ti_{0.87}O_2$ nanosheets and $Li^+$-cations facilitates the migration of Li ions towards the membrane with subsequent diffusion through the membrane. Similar phenomenon was also recently reported in other research works[62,70], in which $Li^+$ ions were observed to pass through the open channels of $TaO_3$ nanosheets with a mesh structure. The Ti vacancies further provide an expressway for rapid transportation of $Li^+$ ions in addition to the conventional interlayer galleries between the $Ti_{0.87}O_2$ sheets[62]. Besides these effects, the nanometric scale of the $Ti_{0.87}O_2$ layers is also favorable for fast Li-ion diffusion.

**Alkali metal anodes with favorable deposition morphology.** Benefiting from the merits mentioned above, $Ti_{0.87}O_2$/PP separators are promising for regulating alkali metal ion flux in electrolyte and facilitating homogenous alkali metal deposition. Asymmetric Li| |Cu cells with various separators were fabricated to evaluate the cycling performance of Li metal anodes during repeated deposition and stripping. As shown in Fig. 3a, the cell with the $Ti_{0.87}O_2$/PP separator exhibited a steady Coulombic efficiency above 96.5% with stable plating/stripping voltage profiles for more than 100 cycles (Fig. 3b). In contrast, the cells with the bare PP (Fig. 3c), anatase $TiO_2$/PP (Supplementary Fig. 24) and GO/PP separators (Supplementary Fig. 25) displayed a gradually increased voltage hysteresis and severely fluctuating Coulombic efficiency, which can be ascribed to the non-uniform Li deposition, and the formation of mossy or dendritic Li on the surface of Li metal anodes. Symmetric Li| |Li cells were assembled to further investigate the superiority of $Ti_{0.87}O_2$/PP separators for stabilizing Li metal anodes. As shown in Fig. 3d, the cell with the $Ti_{0.87}O_2$/PP separator delivered an extended cyclability with stable voltage plateaus (Fig. 3e–g) for over 300 h at a current density of 2 mA cm$^{-2}$ with an area capacity of 1 mAh cm$^{-2}$. In sharp contrast, the cell with the PP separator exhibited a gradual increase in voltage hysteresis (Fig. 3d–f). A similar phenomenon was found for regulating Na deposition and suppressing Na dendrite growth using $Ti_{0.87}O_2$/PP separators. Supplementary Fig. 26 shows the Coulombic efficiencies of asymmetric Na| |Cu cells with PP and $Ti_{0.87}O_2$/PP separators. The corresponding voltage profiles of Na plating/stripping in Na| |Cu half cells with PP and $Ti_{0.87}O_2$/PP separators are shown in Supplementary Figs. 27 and 28, respectively. The average Coulombic efficiency of the cell with the $Ti_{0.87}O_2$/PP separator is about 98.8% for 200 cycles. In contrast, the Coulombic efficiency of the cells with the bare PP decreased below 91% in 150 cycles.

The morphology of the cycled Li anodes in symmetric cells was studied to clarify the effect of the $Ti_{0.87}O_2$ nanosheets on the possible suppression of Li dendrite formation. As shown in Supplementary Figs. 29 and 30, the loosely-stacked mossy Li with a highly porous structure has been observed on the Li anodes from cells with bare PP separators. In contrast, when the $Ti_{0.87}O_2$/PP separator was used, the surfaces of the Li metal anodes were still compact without obvious mossy Li (Supplementary Figs. 31 and 32). This result demonstrates that the $Ti_{0.87}O_2$ nanosheets can facilitate homogeneous $Li^+$ ion flux, giving rise to uniform Li deposition. The cycled $Ti_{0.87}O_2$/PP separator in symmetric cells was also examined via ex situ XPS measurements (Supplementary Fig. 33). Two characteristic peaks of $Ti^{4+}$ without obvious evidence of $Ti^{3+}$ or metallic Ti were observed, indicating a reversible $Li^+$/$Na^+$ diffusion process through the $Ti_{0.87}O_2$/PP separators without the formation of lithium/sodium oxides and other reaction products. Additionally, AFM Young's modulus mappings revealed that $Ti_{0.87}O_2$/PP separators exhibited a modest modulus of around 60 MPa, (Supplementary Fig. 34, see the Experimental details in the Supplementary Information), meeting the requirement for suppressing the growth of Li dendrites[71].

Theoretical calculations were carried out to investigate the diffusion properties of $Li^+$ ions through anatase $TiO_2$ (Fig. 4a), lepidocrocite-type $TiO_2$ without Ti vacancies (Fig. 4b) and Ti-defect-containing $Ti_{0.87}O_2$ (Fig. 4c). Figure 4d shows the transfer profiles of single $Li^+$ ions passing through these layers. For anatase $TiO_2$ and lepidocrocite $TiO_2$, potential-energy barriers are as high as 4.83 and 7.06 eV, respectively. This indicates that it would be challenging for a $Li^+$ ion to diffuse through them. After introducing a Ti vacancy, the energy barrier of the $Ti_{0.87}O_2$ monolayer decreased to 0.75 eV, which is comparable to, or even lower than, that of defective graphene monolayer[72]. Besides these lattice averages, the electronic structure of $Ti_{0.87}O_2$ might also induce lowered energy barriers. The charge density distribution on a $Ti_{0.87}O_2$ lattice with a single Ti cationic defect is shown in Fig. 4e. It can be seen that the charge density around the Ti vacancy can significantly increase the charge attraction for a $Li^+$ ion, reducing the electrostatic charge overlapping, and weakening any Coulombic repulsion between a $Li^+$ ion and the $Ti_{0.87}O_2$ lattice, thus resulting in a lower diffusion barrier for $Li^+$ ions. To further visualize the effects of defective nanosheets on the Li-ion transportation process, two kinds of thin-layer models were constructed by restacking the conventional nanosheets and defective nanosheets, respectively (Supplementary Fig. 35). In the case of the restacked thin layer of conventional nanosheets, the gaps between the adjacent nanosheets were the only pathways for $Li^+$ ion transport. Thus, a non-uniform distribution of $Li^+$ ions was formed (Fig. 4f). In contrast, in the restacked thin layer of cation-defect nanosheets, the $Li^+$ ions could be uniformly redistributed. This can be explained by the fact that $Li^+$ ions migrate through not only the gaps between layers but also the defects within individual layers, resulting in a uniform distribution of Li-ion flux (Fig. 4g). Although the above-idealized models cannot fully reflect all the aspects of real circumstances (especially once electrolyte interactions are introduced into the scenarios), the theoretical calculation and simulation results support the assumption of Li-ion transport promoted by the use of cation-defect nanosheets.

We propose a possible mechanism for the stripping/plating of alkali metal (Li or Na) electrodes when cycled using the PP separator coated with negatively charged $Ti_{0.87}O_2$ nanosheets with atomic Ti vacancies (Supplementary Fig. 36). Upon transport through the coated separator, solvated $Li^+$ ions in liquid electrolyte diffuse to the alkali metal electrode side. The negatively charged $Ti_{0.87}O_2$ nanosheets could attract numbers of $Li^+$ ions and facilitate the de-solvation process of the solvated $Li^+$ ions before deposition, leading to a small energy barrier for deposition[73,74]. Then, the desolvated $Li^+$ ions diffuse through the Ti atomic vacancies. Given the homogenized Ti atomic vacancies of the as-prepared $Ti_{0.87}O_2$ nanosheets, a uniform $Li^+$ flux has been achieved. Consequently, smooth morphologies are obtained on the surface of the alkali metal electrode. However, in the absence of $Ti_{0.87}O_2$ layers, a large energy barrier is needed during the de-solvation process[73,74]. The distribution and transport of $Li^+$ ions are inhomogeneous and then form irregular Li metal depositions (e.g., tips). Subsequently, $Li^+$ ions tend to accumulate at preferentially formed Li irregular deposition, possibly leading to the formation of dendritic structures (Supplementary Fig. 37).

**Reduction of the polysulfide/polyselenide shuttling effect.** In addition to ion re-distribution for a uniform alkali metal deposition, negatively charged $Ti_{0.87}O_2$ can also act as a protective barrier to inhibit the shuttle effect of PS anions. Taking polysulfides as an example, permeation measurements were conducted to evaluate the permeation resistance of $Ti_{0.87}O_2$/PP separators for minimizing the diffusion of PS anions (see the

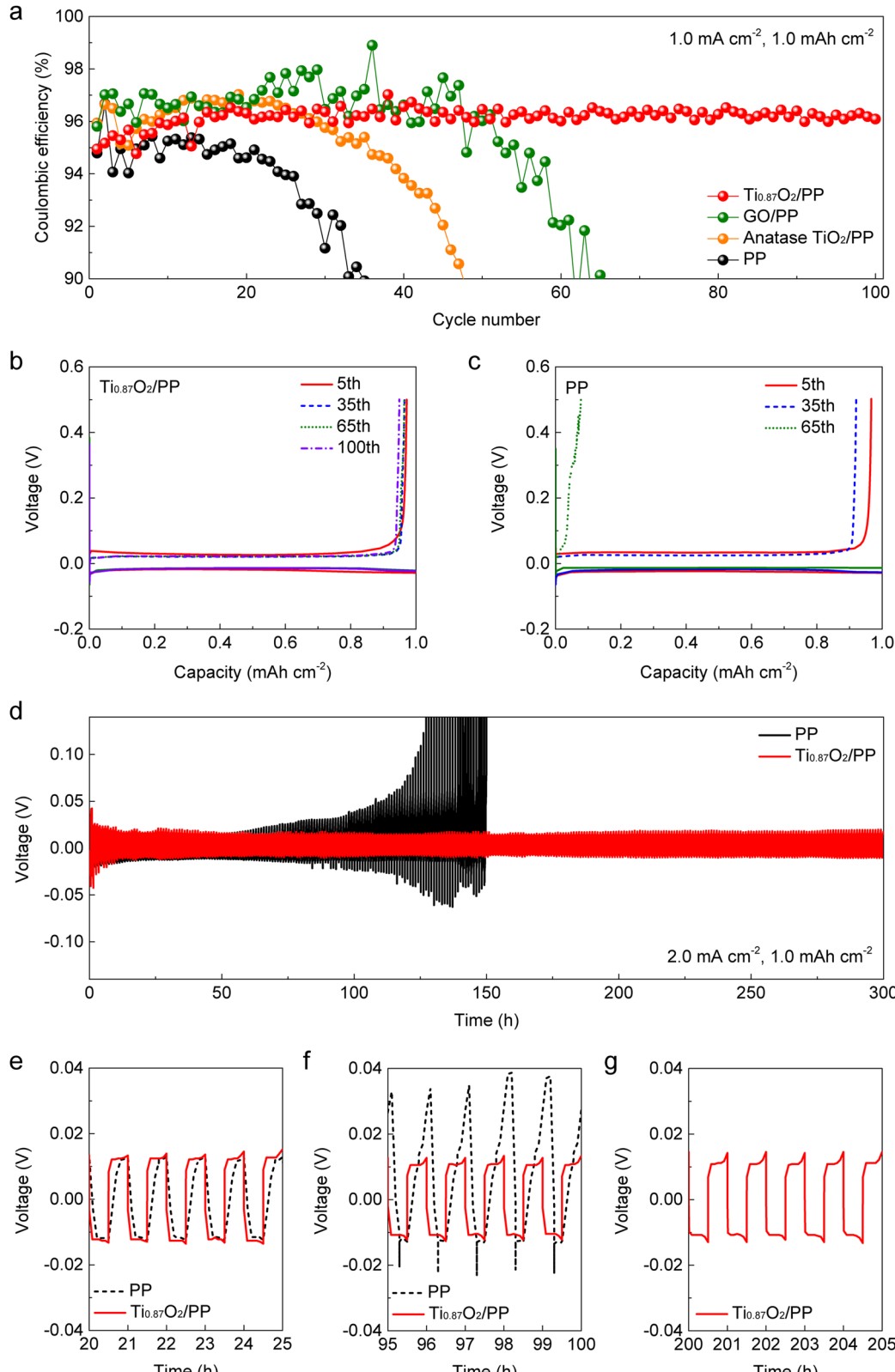

**Fig. 3 Electrochemical performance of asymmetric and symmetric Li metal cells using various separators. a** Coulombic efficiencies of Li| |Cu cells with PP, anatase $TiO_2$/PP, GO/PP, and $Ti_{0.87}O_2$/PP separators with an area capacity of 1 mAh cm$^{-2}$ at 1 mA cm$^{-2}$. Voltage profiles of Li plating/stripping processes in Li| |Cu cells with (**b**) $Ti_{0.87}O_2$/PP and (**c**) PP separators with an area capacity of 1 mAh cm$^{-2}$ at 1 mA cm$^{-2}$. **d** Voltage-time profiles of Li plating/stripping processes in Li| |Li cells with PP and $Ti_{0.87}O_2$/PP separators with selected voltage-time profiles for the (**e**) 21–25 h, (**f**) 96–100 h, and (**g**) 201–205 h.

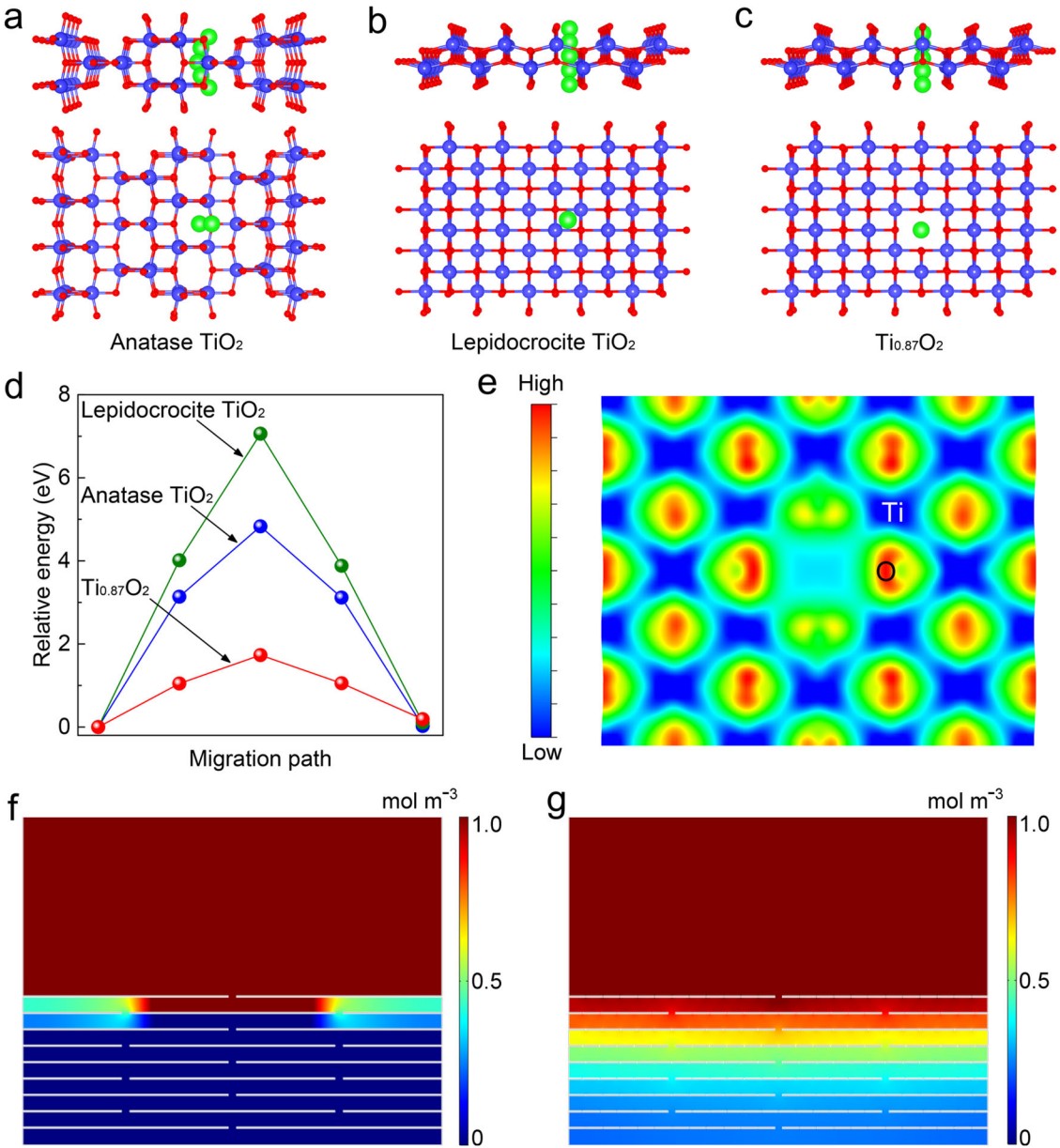

**Fig. 4 Theoretical calculations and simulation results of ion transportation behaviors in $Ti_{0.87}O_2$ nanosheets.** $Li^+$ ion diffusion in (**a**) anatase $TiO_2$, (**b**) lepidocrocite $TiO_2$ sheet, and (**c**) Ti-defective $Ti_{0.87}O_2$ sheet. **d** Potential-energy curves of $Li^+$ ion diffusion in anatase $TiO_2$, lepidocrocite $TiO_2$, and $Ti_{0.87}O_2$. **e** Charge density plot of $Ti_{0.87}O_2$ with a Ti defect. Distribution of $Li^+$ ions through a restacked thin layer of (**f**) conventional nanosheets and (**g**) Ti-defect nanosheets.

Experimental details in the Supplementary Information). Only the $Ti_{0.87}O_2$/PP separator demonstrated a stable blocking effect towards PSs, lasting up to 10 h (Fig. 5a). The diffusion of $Li_2S_6$ was observed when use PP (Fig. 5b) and anatase $TiO_2$/PP separators (Supplementary Fig. 38a) within 1 h. The GO/PP separator was able to suppress the diffusion of PSs during the initial 1 h. However, as time elapsed, PSs were still able to pass through the GO/PP separator (Supplementary Fig. 38b). Both GO and the $Ti_{0.87}O_2$ nanosheets are negatively charged and thus could suppress the shuttling of the negatively charged PS anions via electrostatic repulsion. The different capabilities of GO and $Ti_{0.87}O_2$ for preventing the shuttling of PS anions should be ascribed to their negative charge densities. Based on theoretical calculations (Supplementary Fig. 39), $Ti_{0.87}O_2$ nanosheets have a negative charge density of $1.46\,C\,m^{-2}$, which is over 20 times higher than that of GO ($0.064\,C\,m^{-2}$)[75]. Therefore, the $Ti_{0.87}O_2$

nanosheets with a much higher negative charge density can more effectively inhibit PS shuttling than GO layers. DFT calculations were performed to further elucidate the electrostatic repulsion between PS anions and $Ti_{0.87}O_2$ nanosheets (Fig. 5c–f). Similar calculation methods were also conducted on anatase $TiO_2$ and GO sheets (Supplementary Figs. 40 and 41). As shown in Fig. 5g, the $Ti_{0.87}O_2$ displayed much higher repulsion energies than anatase $TiO_2$ or GO for all PS species.

Ex situ Raman spectroscopy was measured to gain further insights into the suppression of PS shuttling by the $Ti_{0.87}O_2$ nanosheets. Li-S coin cells were disassembled at a given voltage during the charge/discharge processes. We characterized the surfaces of the separators which had been in contact with lithium anodes. Figure 5h, i show the Raman spectra of the PP and $Ti_{0.87}O_2$/PP separators, respectively, retrieved from Li-S cells. For the PP separator (Fig. 5h), three characteristic Raman peaks of

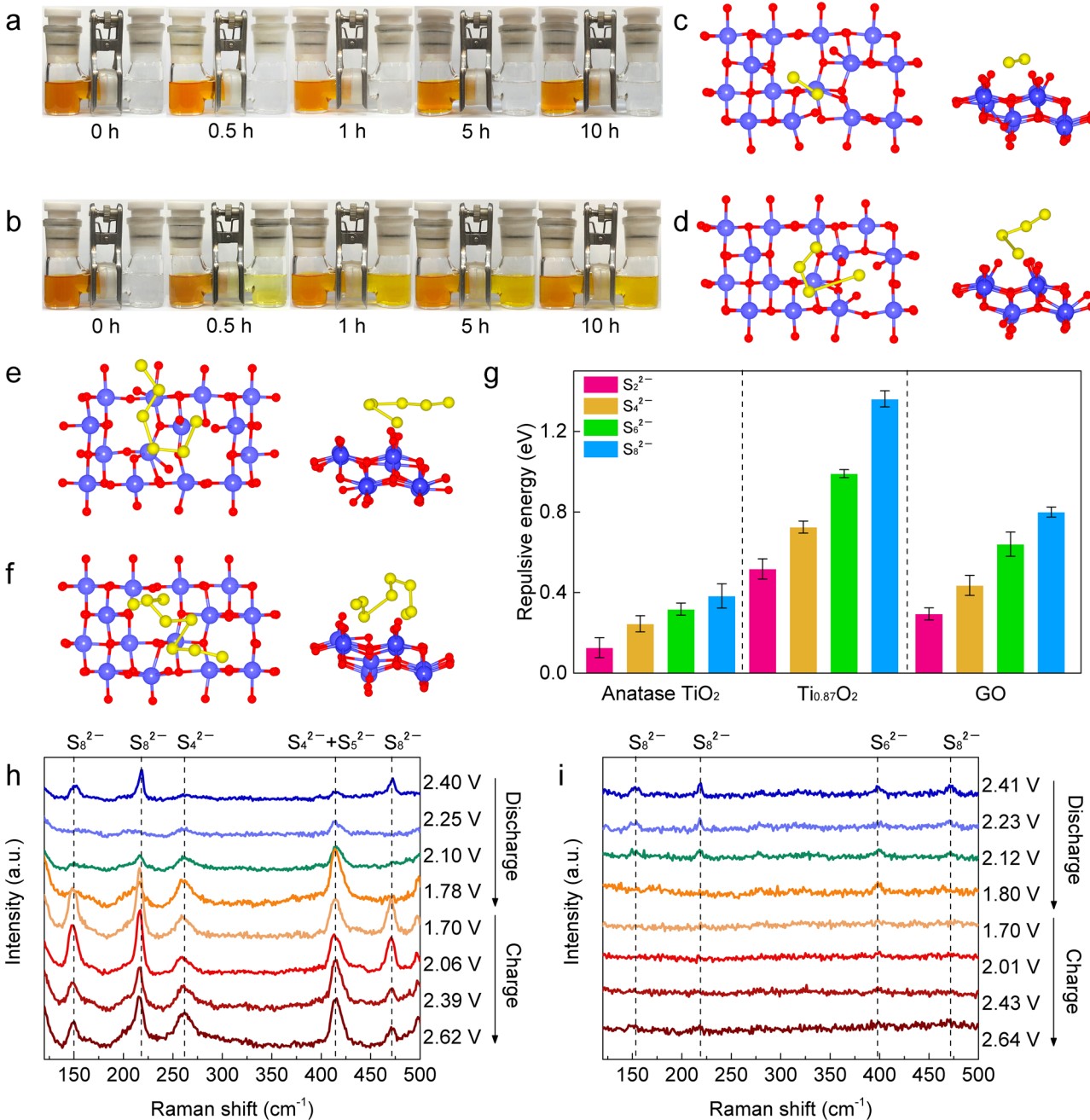

**Fig. 5 Polysulfide shuttling suppression capability for the Ti$_{0.87}$O$_2$/PP separator.** Polysulfide permeation measurements in H-type cells with (**a**) Ti$_{0.87}$O$_2$/PP and (**b**) bare PP separators. Optimized conformations of (**c**) S$_2^{2-}$, (**d**) S$_4^{2-}$, (**e**) S$_6^{2-}$ and (**f**) S$_8^{2-}$ on the Ti$_{0.87}$O$_2$ sheet. **g** Repulsion energies of various polysulfide S$_x^{2-}$ on anatase TiO$_2$, GO, and Ti$_{0.87}$O$_2$. Error bars were included, which represent the standard deviation of the data taken from five optimized configurations. Raman spectra obtained for ex situ samples harvested from cycled cells during the discharge and charge processes in Li-S cells with (**h**) PP and (**i**) Ti$_{0.87}$O$_2$/PP separators.

S$_8^{2-}$ (at ~150, 220, and 470 cm$^{-1}$) were observed in the initial stage of the discharge process, associated with the formation of long-chain PSs. The Raman peaks of S$_8^{2-}$ gradually decreased as the discharge reaction proceeded. Meanwhile, Raman peaks at ~260 and 415 cm$^{-1}$ emerged, which correspond to the short-chain PSs of S$_4^{2-}$ and S$_5^{2-}$. At the end of the discharge process, strong characteristic peaks of S$_4^{2-}$ and S$_5^{2-}$ were observed. This clearly indicates that the PSs shuttled through the PP separator from the cathode side and then deposited on the PP separator facing the anode side. Similarly, during the charging process, strong Raman signals of various PSs were observed. In contrast, for the Ti$_{0.87}$O$_2$/PP separator (Fig. 5i), Raman signals with low intensity of PS species were detected throughout the entire discharge and charge processes, indicating effective reduction of PS shuttling. To observe the reduction of PS shuttling and stabilization of Li metal anodes, the cycled cells were disassembled and the sides of Li metal anodes facing the separators were checked via visual inspection in the Ar-filled glovebox. As shown in Supplementary Fig. 42a, yellow polysulfides were observed on the Li anodes in the cells with PP separators. For the cell with Ti$_{0.87}$O$_2$/PP separators, almost no yellow species were observed and the cycled Li metal still exhibited a bright metallic lustre (Supplementary Fig. 42b). All these results confirmed that the PS shuttling effect and the growth of irregular Li metal

depositions have been effectively reduced when using $Ti_{0.87}O_2$/PP separators. A molecular dynamic simulation further confirmed the decrease of the PS shuttling and regulation of Li-ion transport through the Ti vacancies (Supplementary Movie 1).

**Electrochemical performances.** The electrochemical performances of $Ti_{0.87}O_2$/PP separators in Li-S cells were tested using a carbon black/S composite cathode. Typical cyclic voltammogram (CV) curves of a Li-S cell with a $Ti_{0.87}O_2$/PP separator showed distinct reduction/oxidation peaks, which correspond to the conversion reactions of sulfur cathodes (Supplementary Fig. 43). Li-S cells with different separators were charged and discharged at 0.2 C (1 C = 1675 mA $g^{-1}$). The voltage plateaus of the Li-S cell with a $Ti_{0.87}O_2$/PP separator (Supplementary Fig. 44) were consistent with its CV measurement. The initial discharge capacity was calculated to be 960 mAh $g^{-1}$, followed by a moderate drop to 750 mAh $g^{-1}$ by the end of the 500th cycle (Fig. 6a). In contrast, a cell with a PP separator displayed an initial capacity of 980 mAh $g^{-1}$ (Supplementary Fig. 45) and rapidly decreased to 345 mAh $g^{-1}$ after 500 cycles. For the cells with the anatase $TiO_2$/PP (Supplementary Fig. 46) and GO/PP (Supplementary Fig. 47) separators, lower specific capacities of 450 and 580 mAh $g^{-1}$ were obtained by the end of the 500th cycles, respectively. Supplementary Fig. 48 shows the rate performance of Li-S cells with $Ti_{0.87}O_2$/PP and PP separators at different current rates from 0.2 C to 2 C. The charge-discharge profiles of cells with the $Ti_{0.87}O_2$/PP separators showed distinguishable voltage plateaus at each C rates (Supplementary Fig. 49). High specific capacities of 960 and 560 mAh $g^{-1}$ were achieved at 0.2 C and 2 C, respectively. However, the cells with PP separators suffered from dramatic capacity decay. The capacity reached up to 950 mAh $g^{-1}$ at 0.2 C. However, as the current rate was increased to 2 C, the capacity dramatically decreased to 260 mAh $g^{-1}$. A long-term cycle test was conducted at a 1 C rate for over 5000 cycles to verify the favorable effect of the $Ti_{0.87}O_2$-coated separator on the electrochemical energy storage performances of the cell (Fig. 6b and Supplementary Fig. 50). A specific capacity of 585 mAh $g^{-1}$ was maintained at the end of the 5000th cycle, corresponding to a capacity decay of 0.0036% per cycle. Ex situ SEM images (Supplementary Fig. 51) showed that $Ti_{0.87}O_2$ nanosheets were still maintained on the separator after such long-term cycling. This cycling stability is comparable or even better than the reported functionalized separators for Li-S batteries (Fig. 6c and Supplementary Table 1), including GO[31], graphene[32], G@PC[34], rGO@SL[35], CNT/NCQD[36], MgAl-LDH[44], NiFe-LDH/N-graphene[43], $MoS_2$-PDDA/PAA[40], $Sb_2Se_{3-x}$/rGO[23], $Ti_3C_2$[42], $Cu_2$(CuTCPP)[47], CNT/ZIF-8[46], Ce-MOF/CNT[48], BC/2D MOF-Co[49], and Laponite nanosheets[19].

To explore the potential for practical applications, thick cathodes with a sulfur loading of 3.5 mg $cm^{-2}$ were assembled and examined. Figure 6d and Supplementary Fig. 52 show the long-term cycling performance of a Li-S cell with the $Ti_{0.87}O_2$/PP separator. An initial activation cycles are necessary in Li-S cells[17]. Here, an initial activation process of ~100 cycles at 0.2 C was adopted. The main reason for the capacity decay during the initial activation may result from the dissolution of PSs into the electrolytes and poor electrochemical contacts between the electrolyte and the sulfur particles contained in the cathode composite material during the initial cycles[76,77]. After an initial activation at 0.2 C, the cell delivered a specific capacity of 565 mAh $g^{-1}$ at 1 C up to 5000 cycles. Even at a current rate of 2 C, this cell still delivered a reversible specific capacity of 250 mAh $g^{-1}$ after 10000 cycles, corresponding to a capacity decay as low as 0.0035% per cycle. It should be noted that, to highlight the function of $Ti_{0.87}O_2$ nanosheets, the cathode matrix

of carbon black has almost no PS adsorption ability. The use of porous carbon with hierarchical nanostructures as sulfur cathodes could further increase the sulfur mass loading for higher energy densities. For example, we used commercial carbon nanotubes (CNT) as the sulfur host (Supplementary Fig. 53). The Li-S cells achieved high specific capacities and high areal capacities at various sulfur loading (Supplementary Figs. S54 and S55). Flexible single-layer Li-S pouch cells (6.0 cm × 6.5 cm) were assembled using $Ti_{0.87}O_2$/PP separators. During charging and discharging at different bending angles, the pouch cells exhibited stable cycling performance at a C rate of 0.2 C up to 120 cycles (Fig. 6e and Supplementary Fig. 56).

The applications of $Ti_{0.87}O_2$/PP separators were also extended for Li-Se batteries. Supplementary Figs. 57 and 58 show the typical charge/discharge profiles of Li-Se cells with PP and $Ti_{0.87}O_2$/PP separators, respectively. A gradually increased voltage polarization was observed for the Li-Se cells with PP separators during the initial cycles, accompanied by an obvious capacity decay. In contrast, the overlapped charge/discharge curves confirmed the cycling stability of the Li-Se cells with $Ti_{0.87}O_2$/PP separators. After continuous cycling at 0.2 C for over 500 cycles, a specific capacity of 460 mAh $g^{-1}$ was still retained (Supplementary Fig. 59). The $Ti_{0.87}O_2$/PP separator is also promising to improve the cycling stability for Na-Se batteries. As shown in Supplementary Figs. 60 and 61, highly overlapped charge/discharge curves were observed for Na-Se cells with $Ti_{0.87}O_2$/PP separators, suggesting better cycling performance compared to the cells with bare PP separators. Upon continuous cycling at 0.2 C, a specific capacity of around 450 mA h $g^{-1}$ was achieved after 250 cycles (Supplementary Fig. 62).

In summary, we have demonstrated that PP separator coated with $Ti_{0.87}O_2$ nanosheets with Ti atomic vacancies can be used as a selective ionic sieve to achieve high permeability for regulating alkali metal (Li and Na) deposition while simultaneously preventing PS shuttling for alkali metal-S and alkali metal-Se batteries. The negatively charged $Ti_{0.87}O_2$ nanosheets showed strong electrostatic attraction and re-configurable adhesion for $Li^+$/$Na^+$ ions which enabled $Li^+$/$Na^+$ ions to transit rapidly. The Ti vacancies appear to act as sub-nanometer pores, providing fast diffusion channels for $Li^+$ or $Na^+$-ions. Therefore, a homogeneous distribution of $Li^+$/$Na^+$ ions was achieved at the alkali metal anode side of test cells, inhibiting the growth of Li/Na dendrites without compromising the fast transport of $Li^+$/$Na^+$ ions. On the cathode side, the negatively charged $Ti_{0.87}O_2$ nanosheets showed strong electrostatic repulsion towards PS anions, resulting in effective suppression of PS shuttling. The $Ti_{0.87}O_2$ nanosheets enabled high-performance Li-S, Li-Se, and Na-Se batteries with long cycle lives. Flexible Li-S pouch cells were assembled, showing stable cycling performance under different bending states. This work highlights a strategy of using 2D nanosheets with atomic defects to achieve tandem control of migration of both cations and anions in alkali metal cells.

## Methods

**Synthesis of stable suspensions of anionic nanosheets.** Colloidal suspension of $Ti_{0.87}O_2$ nanosheets was prepared via exfoliation of the polycrystalline layered titanate crystal[57,58]. Typically, $TiO_2$, $K_2CO_3$, and $Li_2CO_3$ in a molar ratio of 1.73: 0.4: 0.14 were mixed and calcinated at a high temperature of 900 °C for 20 h. The obtained layered titanate ($K_{0.8}Ti_{1.73}Li_{0.27}O_4$) was then stirred in a 0.5 M HCl solution at room temperature for 48 h. The acid-exchanged titanate ($H_{1.07}Ti_{1.73}O_4 \cdot H_2O$) was collected by filtration, washed with a copious quantity of deionised water (2–3 L), and air dried at room temperature. Subsequently, the protonic titanate was treated by shaking in a tetrabutylammonium ($TBA^+$) hydroxide aqueous solution. The concentration of $TBA^+$ was 1:1 in molar ratio with respect to the exchanged protons in the titanate. After 7 days, a stable suspension of $Ti_{0.87}O_2$ nanosheets was obtained. For comparison, the suspension of anatase $TiO_2$ nanoparticles was prepared by dispersion of commercial anatase $TiO_2$

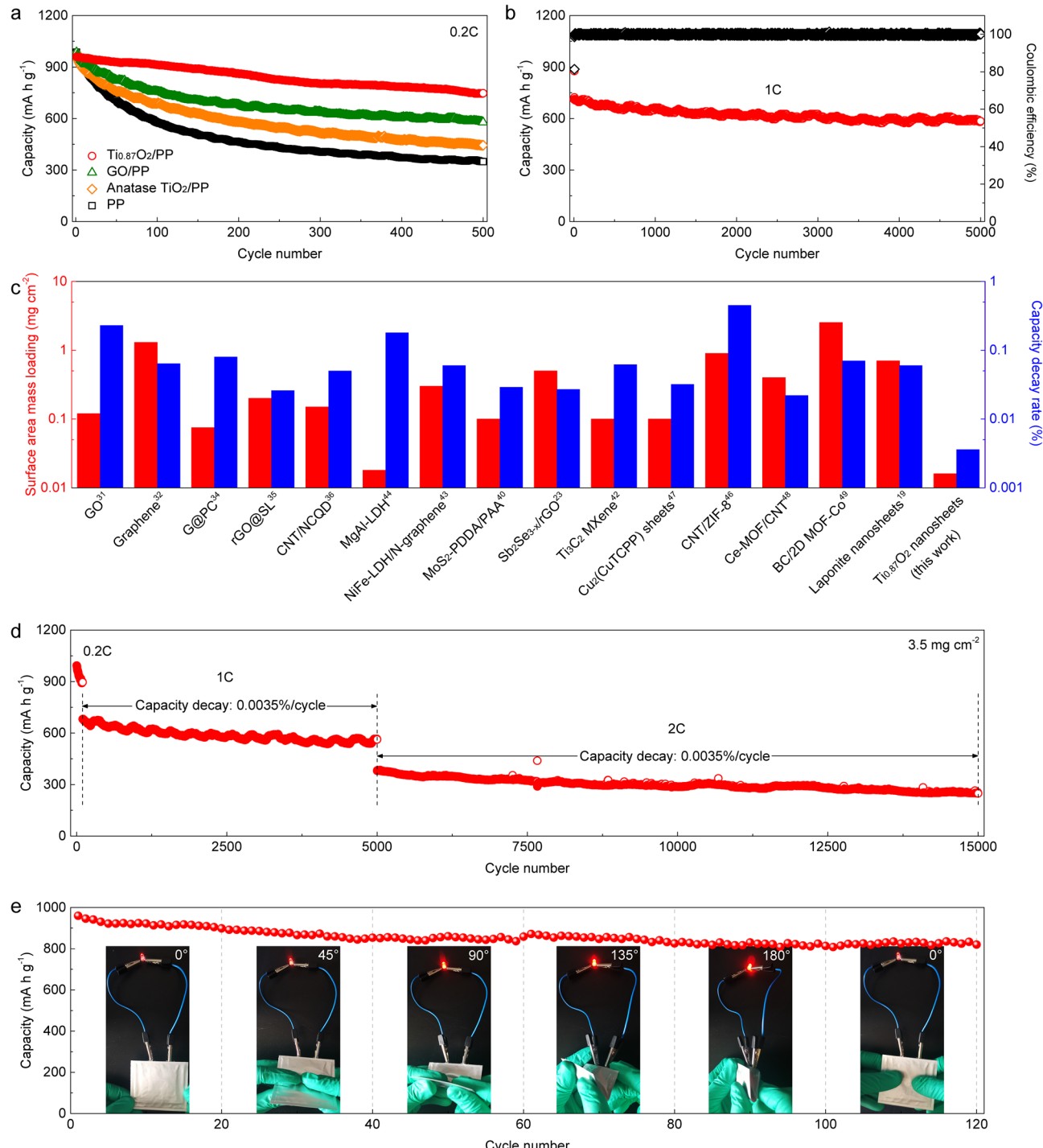

**Fig. 6 Li-S cell performance with Ti$_{0.87}$O$_2$/PP separators. a** Cycling performance with PP, anatase TiO$_2$/PP, GO/PP and Ti$_{0.87}$O$_2$/PP separators at 0.2 C for 500 cycles. **b** Long-term cycling stability of a Li-S cell with the Ti$_{0.87}$O$_2$/PP separator at 1 C for 5000 cycles. **c** Comparison of surface area mass loading and cycling performance for Ti$_{0.87}$O$_2$ nanosheets and some other reported functional layers on commercial separators in Li-S cells. Further details of the selected functional materials are provided in the Supplementary Table S1. **d** Cycling performance of a Li-S cell with the Ti$_{0.87}$O$_2$/PP separator at a sulfur mass loading of 3.5 mg cm$^{-2}$. **e** Cycling performance of a flexible Li-S pouch cell with the Ti$_{0.87}$O$_2$/PP separator under different bending angles.

powder (25 nm, Sigma-Aldrich) in water/ethanol solution. The suspension of graphene oxide (GO) nanosheets was prepared from purified natural graphite by a modified Hummers' method.

**Fabrication of functional separators**. The functional separators were prepared by vacuum filtration of stable suspensions on a commercial PP separator (Clegard 2400). Taking Ti$_{0.87}$O$_2$/PP functional separator as an example, an aqueous suspension of Ti$_{0.87}$O$_2$ nanosheets (0.05 mL, 4 mg mL$^{-1}$) and poly

(tetrafluoroethylene) binder with a mass ratio of 95: 5 were mixed in water/ethanol mixture (v/v = 4: 1). The PTFE was used as a binder to enhance the interfacial adhesion between nanosheets and PP surfaces. After ultrasonication (Elma Ultrasonic Cleaner P120H, 330 W) for 0.5 h, the as-prepared dispersion was vacuum filtered on the commercial PP separators and washed with water/ethanol mixture (Millipore filter with Buchi vacuum pump V-100). The obtained Ti$_{0.87}$O$_2$/ PP separators were dried in vacuum at room temperature (MTI DZF-0620) and then irradiated with UV-light for 2 h (NBET-LED4) before being embedded in electrochemical cells. The illumination with ultraviolet (UV) light in air aims to

photocatalytically decompose the organic ions (such as TBA⁺) surrounding the nanosheets. The surface area mass loading of the $Ti_{0.87}O_2$ nanosheets in the $Ti_{0.87}O_2$/PP separator was estimated to be ~0.016 mg cm⁻². By controlling the volume of the nanosheet suspensions used in the vacuum filtration process, the surface area mass loading of the $Ti_{0.87}O_2$ nanosheets in the $Ti_{0.87}O_2$/PP separator was controlled correspondingly. For comparison, the other functional separators (anatase $TiO_2$/PP and GO/PP) with the same surface area mass loading were prepared according to the same process.

**Characterization**. XRD data were collected on a Bruker D8 Advanced diffractometer at 40 kV, 40 mA for Cu-Kα (λ = 1.5418 Å). The morphologies were studied by a field-emission scanning electron microscope (FE-SEM, Zeiss Supra 55VP) and a JEOL JEM-ARM200F TEM instrument. The zeta-potentials of nanosheet suspensions were determined using an ELS-Z zeta-potential analyzer. AFM measurements were performed on a Dimension 3100 SPM instrument to examine the topography of the nanosheets deposited onto Si wafer substrates. Force curves of the separators are obtained in the force spectroscopy mode using an sQube SiO₂ colloidal probe at a tip velocity of 500 nm s⁻¹ and a trigger point of 0.5 V. Contact angle measurements were conducted by using a KRUSS DSA100 machine. The XANES and EXAFS at Ti K-edge were recorded in transition mode at beamline Spring-8 at the Japan Synchrotron Radiation Research Institute (JASRI). Commercial rutile $TiO_2$ (Sigma-Aldrich) was used as the reference sample. For the ex situ Raman tests, the Li-S coin cells were disassembled in an Ar-filled glove box. The collected separators were washed in 2 mL dimethyl carbonate (DMC), left to dry for 2–3 h in the Ar-filled glove box with water and oxygen levels <0.1 ppm, sandwiched between two glass objective slides and sealed with epoxy resin. A Raman microscope (Labram Aramis, Japan) with a He-Ne laser (532 nm) was used to investigate the side of separators facing the lithium anode.

**Electrochemical measurements**
*Electrochemical testing of Li-S, Li-Se, and Na-Se cells*. Carbon black and carbon nanotube (CNT) were used as host materials for the synthesis of S and Se cathodes. The commercial carbon black powders (Timcal SuperP) and sublimed sulfur (Sigma-Aldrich) (w/w = 3: 7) were mixed and sealed in a glass bottle. After heating at 155 °C for 6 h, the carbon black/S composite was obtained. The commercial CNT powders (Aladdin, China) and sublimed sulfur (w/w = 1: 4) were mixed and sealed in a glass bottle. After heating at 155 °C for 6 h, the CNT/S composite was obtained. The commercial carbon black powders and selenium (Sigma-Aldrich) (w/w = 1: 3) were mixed. After heating at 260 °C for 12 h under N₂ atmosphere in a tube furnace, the carbon black/Se composite was obtained. To prepare the S or Se cathodes, the above S or Se composites, carbon black powders, and poly(vinylidene difluoride) (PVDF) binder were mixed with a mass ratio of 80: 15: 5 in N-methyl-2-pyrrolidinone (NMP). Then the mixture slurry was coated on aluminum foil and dried at 60 °C under vacuum (MTI DZF-0620). The mass loading of sulfur was controlled to be ~1.5 mg cm⁻² for regular tests and ~3.5, 6.1 and 8.9 mg cm⁻² for the high-sulfur-loading tests. Coin type (CR2032) cells were fabricated in an argon-filled glovebox using S/Se composite cathodes, metallic Li/Na anodes and functional separators. The lithium metal disk (purity: 99.9%, diameter: 16 mm, thickness: 0.6 mm) was obtained from SCI Materials Hub. The sodium pieces (in kerosene, ≥99.8%, Sigma-Aldrich) were cut to the appropriate size. For the Li-S and Li-Se cells, the electrolyte was 1 M bis(tri-fluoromethane) sulfonimide lithium (LiTFSI) in a mixed solvent of 1,2-dimethoxyethane (DME) and 1,3-dioxacyclopentane (DOL) (1:1, v/v) with LiNO₃ (1 wt%). For the Na-Se cells, the electrolyte was 1 M sodium perchlorate (NaClO₄) in ethylene carbonate (EC) and diethyl carbonate (DEC) (1/1, v/v) with 2 % fluoroethylene carbonate (FEC). All the electrolytes were purchased from DoDoChem with a water content <50 ppm. The CV tests were carried out using a VMP3 electrochemical workstation (Bio-Logic Inc.) in a potential window of 1.7–2.8 V vs. Li/Li⁺ at 0.1 mV s⁻¹. The galvanostatic discharge/charge profiles (1.7–2.8 V) of the batteries were recorded using a LAND CT2001A battery test station (1 C = 1675 mA g⁻¹). All electrochemical tests were carried out in an environmental chamber at a temperature of around 25 °C.

*Lithium-sulfur pouch cell assembly*. The Li-S pouch cells (6.0 cm by 6.5 cm with Al-plastic film as package material) were assembled in an Ar-filled glove box. The CNT/S composite, SuperP carbon black powders, and PVDF binder were mixed with a mass ratio of 80: 15: 5 in NMP. The resulting slurry was casted on the Al foil (4.8 cm by 4.8 cm) and dried at 60 °C in a vacuum oven overnight. The sulfur mass loading of the as-prepared cathode was ~5 mg cm⁻². The Al tab was riveted on the as-prepared cathode, and a Ni tab was riveted on the Cu foil current collector. A piece of Li foil (4.8 cm by 4.8 cm) with a thickness of 0.1 mm was coated onto copper foil collectors by a continuous roller. An electrolyte-soaked $Ti_{0.87}O_2$/PP separator (5.0 cm by 5.0 cm) was stacked on the surface of the Li foil. Then, the cathode was put on the top of the $Ti_{0.87}O_2$/PP separator. Subsequently, 0.4 mL of electrolyte was injected into the stack. Finally, the stacked layers were inserted into vacuum ziploc bag and sealed under vacuum to get a pouch cell.

*Electrochemical testing of lithium plating/stripping*. To investigate the stripping and plating of Li anode, the symmetric Li//Li and asymmetric Li//Cu cells are assembled using different separators. The electrolyte was the 1 M LiTFSI in a

mixture of DOL and DME (1:1, v/v) containing 1 wt% LiNO₃. For the ex situ SEM and XPS tests, the symmetric cells were disassembled in an Ar-filled glove box. The collected Li metal anodes and separators were washed by 2 mL DMC. Before characterizations, the Li metal anodes and separators were dried in the Ar-filled glove box with water and oxygen levels <0.1 ppm. To prevent being oxidized in air, the Li metal anodes were sealed in a quartz dish in Ar and were transferred into the test chamber using a sealed Ar-filled vessel.

*Electrochemical testing of sodium plating/stripping*. Two-electrode coin cells (CR2032) were assembled in an argon-filled glove box with water and oxygen levels less than 0.1 ppm for electrochemical testing. An electrolyte of 1 M sodium triflate in diglyme was prepared in an argon-filled glove box. The water level of the final electrolyte solution was less than 25 ppm determined by a Mettler Toledo C20 Karl Fischer Titrator. The Na plating/stripping study was conducted on Neware(TM) battery testers at room temperature. For the Coulombic efficiency testing, in each galvanostatic cycle, Na was deposited on different current collectors at the desired current density and capacity, and stripped away by charging to a cut-off voltage of 0.5 V vs. Na⁺/Na.

**Polysulfide permeation measurements**. The 0.05 M polysulfide ($Li_2S_6$) solution was prepared by dissolving sublimed sulfur and $Li_2S$ powder (mole ratio = 5: 1) into a mixed solvent of 1,2-dimethoxyethane (DME) and 1,3-dioxacyclopentane (DOL) (1:1, v/v), followed by magnetic stirring at 50 °C for 12 h. Then, the as-prepared $Li_2S_6$ solution was filled in one side of the U-shaped glass bottles. The other chamber was filled with DME/DOL solvent without $Li_2S_6$. These two chambers were separated by the pristine PP separator and the functional separators with the same mass loading.

**Li-ion transference number**. The lithium-ion transference numbers for PP, anatase $TiO_2$/PP, GO/PP, and $Ti_{0.87}O_2$/PP separators were determined with chronoamperometry at a constant step potential of 10 mV. Each separator was separately sandwiched between two lithium metal electrodes in a coin-type cell (CR 2032). The lithium-ion transference number ($t_{Li+}$) was calculated from the ratio of steady-state current ($I_s$) to initial state current ($I_o$) according to the following equation:

$$t_{Li+} = \frac{I_s}{I_0} \tag{1}$$

**Ionic conductivity**. The ionic conductivities of PP, anatase $TiO_2$/PP, GO/PP, and $Ti_{0.87}O_2$/PP separators were calculated from electrochemical impedance spectroscopy (EIS) measurements. The separator saturated with electrolyte was sandwiched between two stainless steel electrodes in coin-type cells (CR 2032). The EIS tests for these symmetric cells were carried out at Open Circuit Potential (OCP) using a VMP3 electrochemical workstation (Bio-Logic Inc.) with an alternating-current (AC) voltage amplitude of 5 mV in a frequency range of 0.01–100 kHz. The bulk resistance was determined by the intercept of Nyquist plot with real axis. The ionic conductivity was calculated according to the following equation:

$$\sigma = \frac{l}{R_b A} \tag{2}$$

where σ stands for ionic conductivity, $l$ represents the thickness of the membrane, $A$ is the area of the stainless steel electrode, and $R_b$ refers to the bulk resistance.

**Young's modulus**. The Young's Modulus of the $Ti_{0.87}O_2$/PP separators was extracted from the force profiles using a Hertzian model. For paraboloidal indenters, the force-indentation relation is given by

$$F = \frac{4E\sqrt{R}}{3(1-v^2)}\delta^{3/2} \tag{3}$$

where $F$ is the applied load, $E$ is the Young's Modulus, $R$ is the tip radius of curvature, $\delta$ is the indentation depth, and $v$ is the Poisson's ratio. Eq. (3) can be further rearranged to fit a linear model of force²/³ vs. indentation depth;

$$F^{2/3} = \left[\frac{4E\sqrt{R}}{3(1-v^2)}\right]^{2/3}\delta \tag{4}$$

The Young's Modulus can therefore be extracted from the slope of the linear regime of a force²/³ vs. indentation plot given by Eq. (4)[78]. The probe diameter is 5 μm. The Poisson's ratio of the $Ti_{0.87}O_2$ is υ = 0.31 ref. [79].

**DFT calculations**. All the calculations were performed using the framework of spin-polarized DFT as implemented in the Vienna Ab initio Simulation Package (VASP)[80]. The exchange-correlation potentials were treated by the generalized gradient approximation (GGA)[81] parameterized by Perdew, Burke, and Ernzerhof (PBE)[82]. The interaction between valence electrons and ion cores was described by the projected augmented wave (PAW) method[83], and the DFT-D2 method considering van der Waals (vdW) interaction was adopted for the adsorption system. The climbing image nudged elastic band (CI-NEB) method implemented in VASP transition state tools is used to determine the metal cationic minimum energy diffusion pathways and

the corresponding energy barriers[84]. In this step, the algorithm to relax the ions into their energy minimization transition state is required in agreement with the previous calculation of initial and final state. The electronic wave functions were expanded in a plane-wave basis with a cutoff energy of 400 eV. We adopted completely the same k-mesh density and convergence accuracy as ion relaxation both in the geometry optimization calculations and transition state calculations. The Brillouin zone (BZ) is sampled with a $3 \times 1 \times 3$ Monkhorst–Pack scheme k-point mesh[85]. The convergence criterion for energy and force was set at $1.0 \times 10^{-4}$ eV/atom and 0.01 eV/Å, respectively[85]. The vacuum space of 20 Å was set to avoid unexpected interactions between atoms in different cells.

**Simulations of Li-ion transportation.** Li-ion transport processes in two thin layers of restacked nanosheets were simulated in COMSOL[86]. The thin layer with a thickness of 80 nm was composed of restacked conventional nanosheets and defective nanosheets. The thickness of a single nanosheet was set as 1 nm. Although most of the Ti vacancies are single vacancies of the octahedral cavity, there are some cluster-like continuous vacancies with larger sizes[61]. For the sake of simplification, the size of the defects on defective nanosheets was set as 0.5 nm. The gap and distance of two adjacent nanosheets were set as 5 and 10 nm, respectively. The thin layers were confined in an electrolyte-filled region as a closed system. Time dependent form of the Nernst–Planck equation was employed to describe the transport process and snapshots of concentration profiles in the two models were taken after a fixed electrochemical reaction time elapsed in the simulation. The electrolyte was set with an initial concentration 1 M, conductivity of $1 \times 10^{-4}$ S m$^{-1}$, and Li-ion diffusion coefficient of $1 \times 10^{-9}$ m$^2$ s$^{-1}$ ref. [87]. Diffusion inside the nanosheets was not considered for simplicity.

**Molecular dynamic simulation.** The model of the Ti$_{0.87}$O$_2$ phase was build using Virtual Molecular Dynamic (VMD) software. A single Ti vacancy was configured in the Ti$_{0.87}$O$_2$ slab according to the chemical formula. The bond and angle parameters for S$_6$ intermolecular interactions were selected based on previous studies[88,89]. The Lennard-Jones non-bonded parameters were selected based on previous reports[90]. The electric field was applied in the Y direction to propel Li$^+$ and S$_6^{2-}$. The molecular dynamic simulation was performed via Nanoscale Molecular Dynamics (NAMD) in periodic boundary conditions via the canonical ensemble (NVT), and the simulation was visualized using VMD software.

## Data availability

All relevant data are available from the corresponding author upon reasonable request.

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

## Acknowledgements

P.X. and J.Z. acknowledge the financial supported by the Natural Science Foundation of China (No. 51902161, 51772152, 11574262) and the Fundamental Research Funds for the Central Universities (No. 30919011269). G.W. acknowledges the financial support by the Australian Research Council (ARC) through the Discovery Project program, (DP200101249). This work was also supported in part by the WPI-MANA, Ministry of Education, Culture, Sports, Science and Technology, Japan and CREST of the Japan Science and Technology Agency (JST).

## Author contributions

P.X., J.Z., and G.W. designed the research. P.X. and F.Z. conducted the synthesis, characterizations, and electrochemical measurements. P.X., X.Z., and J.S. performed the theoretical calculations. Y.L., Y.W., S.W., and B.S. helped the electrochemical measurements. Z.L. helped the microscopy characterizations. P.X., R.M., T.S., J.Z., and G.W. analyzed and discussed the experimental results and drafted the manuscript. B.S., Z.L., Y.B., and X.W. joined the discussion of data and gave useful suggestions. P.X., F.Z., and X.Z. contributed equally to this work.

## Competing interests

The authors declare no competing interests.
