## [Peer Review File · Nature Communications]

REVIEWER COMMENTS

Reviewer #1 (Remarks to the Author):

Authors have synthesised a negatively charged state Ti_{0.87}O₂ nanosheet as a selective ion sieve for alkali metal sulfur/selenium batteries. Electrochemical analysis is carried out using the functionally modified Ti_{0.87}O₂/PP separator for Li-S, Li-Se, Na-Se batteries.

There are few queries the authors should address before the acceptance of this article.

1. Since the Li⁺/Na⁺ pass through the Ti_{0.87}O₂ modified PP separator, wont there be a chance of forming oxides and other reaction products of lithium/sodium? Is there any qualitative analysis authors could provide to eliminate the possibility for the formation of lithium/sodium oxides and other reaction products?
2. Though Ti_{0.87}O₂ nanosheets are unilamellar with uniform thickness, from the AFM image (Figure 2b) these nanosheets above PP appears to have wide cracks, this could allow the shuttling of PS's. The authors are requested to comment on this.
3. The lithium-ion diffusion through the PP separator decreases when the pores in the PP separator is covered. Did the authors optimize on the thickness of Ti_{0.87}O₂ nanosheets over PP separator for high lithium-ion migration?
4. When cycled at 0.2C with 3.5 mg cm⁻² sulfur loading (Figure 6d), what could be the reason for the drop in specific capacity with cycle?.

Reviewer #2 (Remarks to the Author):

This paper describes the preparation and properties of a composite separator membrane consisting of a thin film of a 2 D Ti_{0.87} O₂ nanosheet with sub-nm pores assembled on Celgard 2400, commercially available, monolayer PP membrane. The layers were self-assembled layer by layer using a vacuum filtration method.

Multifunctional approaches to facilitate cationic (Li⁺) and prevent anionic (polysulfide) migration in LiS battery separators are well known using the principles of physisorption, chemisorption and electrostatic repulsion (Gupta and Sivaram, Energy Technol. 2019, 1800819). However, this paper is a step change in the strategy to combine selective sieving (pore-engineering) with electrostatic forces to improve the performance of the separator membrane.

The properties reported for the composite membrane are attractive; no shrinkage at 120°C, improved surface wettability and reduced propensity to dendrite formation. The surface negative charge facilitates Li⁺ transport while providing electrostatic repulsion for inhibiting polysulfide migration and the small pore size also allows only Li⁺ to pass through (ion-sieving effect) The concept of tailoring a separator surface has been well described in the manuscript and the benefits that accrue to the operation of a LiS batteries are supported by adequate data. The superiority of the approach is evident from the Table provided by the authors in the supplementary section. The cycling performance is superior to all known separators in the literature. This work will advance the field of tailored separator surfaces for demanding battery applications with special reference to LiS batteries.

Nevertheless, there are a few issues that the authors should address before the paper can be accepted for publication.

1. The nature of adhesion between the polar Ti_{0.87} O₂ nanosheet and the PP surface has not been addressed. How stable are these surfaces for long term use?
2. The process of deposition uses 5% PTFE and UV irradiation. The purpose of these treatment is not described.
3. Lithium ion conductivity shows only a modest 25 % increase. Why?
4. Celgard 2400 is known to possess wide slit-like macropores. Obviously the nanosheet will penetrate these pores and block the wide pores of the PP. This is evident from an examination of Figures S3 and S6. What would be the consequence of this to the flux of the electrolyte?

5. Celgard 2400 do not possess tortuous pores. Therefore, covering the open pores of Celgard 2400 will not increase the tortuosity as stated in p.12.

6. Maximum lithium ion conductivity is observed at 0.016 mg/cm² weight density. Therefore, it is necessary to show the SEM of this sample, which is missing

Dr. Swaminathan Sivaram

Reviewer #3 (Remarks to the Author):

The manuscript entitled „Atomic-scale tandem regulation of anionic and cationic migration for long-life alkali metal batteries“ deals with modification of the PP separator with negatively charged TiO₂- nanosheets with cation vacancies on the cathodic and anodic side of Li-S and Na-Se batteries in order to support the diffusion of smaller ions like Li⁺/Na⁺ or S₂⁻/Se₂⁻, but to impede the diffusion of polysulfides and polyselenides. The manuscript implies a lot of characterization methods and is very comprehensive.

However, the results obtained in present work are comparable with other results on the field of Li-S batteries in terms of specific capacity, long-term cycling and a sulfur loading.

Thus, the average value of about 650 mAh g⁻¹ at the current density of 1C was obtained for 5000 cycles, while about 350 mAh g⁻¹ at 2C for the next 10000 cycles, with the mass loading of 3.5 mg cm⁻².

According to the literature, comparable characteristics (high mass loading and stable cycling behavior) are already known for Li-S batteries.

For example, G. Zhou et al. reported in 2015 about specific capacity of 500 mAh g⁻¹ at 0.9C after 1000 cycles with a much higher sulfur mass loading of 10.1 mg cm⁻² (Nano Energy 2015, 11, 356) using the graphene foam S-cathode. The graphene foam provided an electrically conductive network, robust mechanical support and sufficient space for a high sulfur loading.

Next, a protective coating of the PP separator by mesoporous carbon layer on the cathodic side allowed localization of dissolved polysulfide intermediates and retained them as active material within the cathode side, suppressing their further diffusion to the anodic side (Adv. Funct. Mater. 2015, 25, 5285). A stable cycling with specific capacity of 900 mAh g⁻¹ at 1C for 500 cycles was observed.

Further, a 2D protective MoS₂ anodic layer, coated on the Li-anode, enables specific capacity of about 1000 mAh g⁻¹ at 0.5C for at least 1200 cycles (Nature Nanotechnology, 2018, 13, 337). A uniform Li-deposition and stripping without dendrite formation and suppression of polysulfides diffusion through the 2D layers to the anode side were the reasons for the observed stable cycling behavior.

Therefore, the results from the current manuscript represent a further improvement of the already existing strategies for enhancement of the Li-S battery performance and are not novel. I believe that due to this reason the manuscript is not suitable for Nat. Comm. and must be transferred to a more technological journal.

Some comments/suggestions

1) In the introduction part, the authors write “At the S/Se cathode side of alkali metal batteries (Figure 1d), the negatively charged TiO₂ nanosheets with a high negative charge density effectively exclude PS anions via a strong electrostatic repulsion effect.”

It would be more correct to write that PS anions cannot go through the TiO₂ nanosheets primary because of the geometrical restrictions. A comparison of the anionic size for sulfide anion S(2-) and for example for polysulfide anion S₄(2-) (approximately 1.84 Å vs. 3.7-4.0 Å) speaks rather for impossible diffusion of polysulfide anions through the defect sites of TiO₂ nanosheets. Moreover, the local negative charge density is higher for S(2-) than for polysulfides, since in PS the charge is distributed within the bigger molecule. Therefore, the electrostatic repulsion must be higher in case of TiO₂ nanosheets and S(2-).

2) Did authors confirm the composition of $\text{Ti}_{0.8}\text{O}$ nanosheets by the chemical analysis? An XRD analysis of $\text{Ti}_{0.8}\text{O}$ nanosheets without PP separator would be useful as well. What about hydrogen content in the nanosheets? According to the stoichiometry of $\text{Ti}_{0.8}\text{O}$, there is a quite large negative charge on the surface. Could authors exclude the formation of $\text{TiO}_2 \cdot x\text{H}_2\text{O}$ with structural water, which cannot be removed during drying process under soft conditions?

Response to Reviewers' Comments

We would like to thank all reviewers for taking time and efforts to review our manuscript. We sincerely appreciate all the reviewers for their valuable comments and suggestions, which helped us to improve the overall quality of the manuscript. Our point-by-point responses to the reviewers' comments are described below. All revised contents and added revisions have been added and highlighted in the revised manuscript and revised supporting information.

Reviewer #1:

Authors have synthesised a negatively charged state $Ti_{0.87}O_2$ nanosheet as a selective ion sieve for alkali metal sulfur/selenium batteries. Electrochemical analysis is carried out using the functionally modified $Ti_{0.87}O_2/PP$ separator for Li-S, Li-Se, Na-Se batteries. There are few queries the authors should address before the acceptance of this article.

Response: We sincerely appreciate the reviewer's valuable comments and suggestions on our work. According to the reviewer's suggestions, we had carefully revised our manuscript.

Comment 1: *Since the Li^+/Na^+ pass through the $Ti_{0.87}O_2$ modified PP separator, wont there be a chance of forming oxides and other reaction products of lithium/sodium? Is there any qualitative analysis authors could provide to eliminate the possibility for the formation of lithium/sodium oxides and other reaction products?*

Response: We appreciate the reviewer's valuable suggestions. The symmetric cells were disassembled and the chemical analysis of the cycled $Ti_{0.87}O_2/PP$ separator was conducted via the X-ray photoelectron spectroscopic (XPS). Figure S33 compared Ti 2p XPS spectra of the pristine and cycled $Ti_{0.87}O_2/PP$ separators disassembled from the symmetrical cell at a current density of 2 mA cm^{-2} with a capacity of 1 mAh cm^{-2} for 20 cycles. Two characteristic peaks of Ti^{4+} without obvious evidence of Ti^{3+} or metallic Ti were observed, indicating no formation of lithium/sodium oxides and other reaction products in the cycled $Ti_{0.87}O_2/PP$ separators. These results support a reversible Li^+/Na^+ diffusion process through the $Ti_{0.87}O_2/PP$ separators without the formation of lithium/sodium oxides and other reaction products. The related discussion has been added in the revised manuscript.

Corresponding revisions in Manuscript Page 15:

The cycled $\text{Ti}_{0.87}\text{O}_2/\text{PP}$ separator in symmetric cells was also examined (Figure S33). Two characteristic peaks of Ti^{4+} without obvious evidence of Ti^{3+} or metallic Ti were observed, indicating a reversible Li^+/Na^+ diffusion process through the $\text{Ti}_{0.87}\text{O}_2/\text{PP}$ separators without the formation of lithium/sodium oxides and other reaction products.

Corresponding revisions in Supplementary Information:

Figure S33. High-resolution spectrum of Ti 2p of pristine and cycled $\text{Ti}_{0.87}\text{O}_2/\text{PP}$ separators disassembled from the symmetrical cell at a current density of 2 mA cm^{-2} with a capacity of 1 mAh cm^{-2} for 20 cycles.

Comment 2: *Though $\text{Ti}_{0.87}\text{O}_2$ nanosheets are unilamellar with uniform thickness, from the AFM image (Figure 2b) these nanosheets above PP appears to have wide cracks, this could allow the shuttling of PS's. The authors are requested to comment on this.*

Response: We apologize for making it confusing here. Figure 2b is the AFM image of $\text{Ti}_{0.87}\text{O}_2$ nanosheets, in which the nanosheets were deposited on Si wafer substrates for the AFM observation. We have clarified this point in the figure caption of Figure 2b in the revised manuscript. The top-view characterization of the $\text{Ti}_{0.87}\text{O}_2/\text{PP}$ separators can be found in Figure 2h, where the surfaces of PP were fully and uniformly covered by the $\text{Ti}_{0.87}\text{O}_2$ nanosheets. Due to the negative charge and small Ti vacancies of the $\text{Ti}_{0.87}\text{O}_2$ nanosheets, the shuttling of PS anions was effectively excluded via a strong electrostatic repulsion effect and a size-sieving effect.

Corresponding revisions in Manuscript Page 11:

Figure 2: (b) AFM image of $\text{Ti}_{0.87}\text{O}_2$ nanosheets. The nanosheets were deposited onto Si wafer substrates.

Comment 3: *The lithium-ion diffusion through the PP separator decreases when the pores in the PP separator is covered. Did the authors optimize on the thickness of $\text{Ti}_{0.87}\text{O}_2$ nanosheets over PP separator for high lithium-ion migration?*

Response: We appreciate the reviewer's kind reminder. The $\text{Ti}_{0.87}\text{O}_2/\text{PP}$ separators had been prepared with various weight densities and thicknesses. The related XRD and SEM characterizations of the $\text{Ti}_{0.87}\text{O}_2/\text{PP}$ separators with various weight densities have been shown in Figures S7-S11. Figure S7 shows X-ray diffraction (XRD) data of $\text{Ti}_{0.87}\text{O}_2/\text{PP}$ separators with different weight densities. As the weight density increased, the XRD peaks became more intense, with increasing thickness of the $\text{Ti}_{0.87}\text{O}_2$ layer. The morphology and cross-sectional characteristics of $\text{Ti}_{0.87}\text{O}_2/\text{PP}$ separators with relatively high weight densities were also investigated (Figures S8-S11). As the reviewer mentioned, covering open pores of pristine separators will increase the tortuosity of ion movement, leading to reduced Li-ion diffusion. According to the reviewer's comment, the thickness of $\text{Ti}_{0.87}\text{O}_2$ nanosheets over PP separator has been optimized for high lithium-ion migration. As shown in Figures S21 and S23, the Li ion conductivity and Li ion transference number of the $\text{Ti}_{0.87}\text{O}_2/\text{PP}$ separator decreased when the weight density and thickness increased. An $\text{Ti}_{0.87}\text{O}_2/\text{PP}$ separator with a weight density of 0.016 mg cm^{-2} was optimized with the largest Li ion conductivity and Li ion transference number. The related discussion has been added in the revised manuscript.

Corresponding revisions in Manuscript Page 13:

As shown in Figures S21 and S23, the Li ion conductivity and Li ion transference number of the $\text{Ti}_{0.87}\text{O}_2/\text{PP}$ separator decreased when the weight density and thickness increased. An optimized $\text{Ti}_{0.87}\text{O}_2/\text{PP}$ separator with a weight density of 0.016 mg cm^{-2} was achieved with the largest Li ion conductivity and Li ion transference number.

Corresponding revisions in Supplementary Information:

Figure S21. Li ion conductivity of PP, anatase TiO_2/PP , GO/PP and $\text{Ti}_{0.87}\text{O}_2/\text{PP}$ separators with different weight densities.

Figure S23. Li ion transference number of PP, anatase TiO_2/PP , GO/PP and $\text{Ti}_{0.87}\text{O}_2/\text{PP}$ separators with different weight densities.

Comment 4: When cycled at 0.2C with 3.5 mg cm^{-2} sulfur loading (Figure 6d), what could be the reason for the drop in specific capacity with cycle?

Response: We apologize for not giving a clear explanation in the previous manuscript. The drop of specific capacity during the initial 100 cycles at 0.2C is an initial activation, which is widely observed in previous reports [*Nat. Energy* **1**, 16094 (2016); *Adv. Mater.* **30**, 1706895 (2018); *Adv.*

Energy Mater. **6**, 1502539 (2016); *Angew. Chem. Int. Ed.* **57**, 16703–16707 (2018); *J. Mater. Chem. A* **2**, 15889–15896 (2014); *Proc. Natl. Acad. Sci. U.S.A.* **117**, 14712–14720 (2020); *ACS Energy Lett.* **5**, 2234–2245 (2020)]. The main reason for the capacity decay during the initial activation may result from the dissolution of PSs into the electrolytes and poor electrochemical contacts between the electrolyte and the internal sulfur particles during the initial cycles [*J. Mater. Chem. A* **2**, 15889–15896 (2014); *ACS Energy Lett.* **5**, 2234–2245 (2020)]. It should be noted that, to highlight the function of Ti_{0.87}O₂ nanosheets, the carbon black was used as the cathode matrix, which has limited PS adsorption ability. Therefore, a capacity decay during the initial activation was observed when cycled at 0.2C with 3.5 mg cm⁻² sulfur loading (Figure 6d). The initial activation cycles are necessary in Li-S batteries due to the large interfacial contact area between carbon black hosts and sulfur. When cycled at an increased sulfur loading, intimate interconnections among sulfur particles play a significant role in improving the ionic transportation inside the carbon/sulfur composite cathodes [*Nat. Energy* **1**, 16094 (2016); *J. Phys. Chem. Lett.* **5**, 915–918 (2014); *Electrochim. Acta* **56**, 9549–9555 (2011)]. Thus, in our case, an initial process of approximately 100 cycles at 0.2C was first measured, after which a highly reversible and stable electrochemical status could be achieved. The related discussion has been added in the revised manuscript.

Corresponding revisions in Manuscript Page 24:

An initial activation cycles are necessary in Li-S batteries¹⁷. Here, an initial activation process of approximately 100 cycles was first measured. The main reason for the capacity decay during the initial activation may result from the dissolution of PSs into the electrolytes and poor electrochemical contacts between the electrolyte and the internal sulfur particles during the initial cycles^{76,77}.

Corresponding revision in References:

[17] Bai, S. *et al.* Metal-organic framework-based separator for lithium-sulfur batteries. *Nat. Energy* **1**, 16094 (2016).

[76] Zhang, S. *et al.* Activated carbon with ultrahigh specific surface area synthesized from natural plant material for lithium–sulfur batteries. *J. Mater. Chem. A* **2**, 15889–15896 (2014).

[77] Ye, H. *et al.* Activating Li₂S as the lithium-containing cathode in lithium–sulfur batteries. *ACS Energy Lett.* **5**, 2234–2245 (2020).

Reviewer #2:

This paper describes the preparation and properties of a composite separator membrane consisting of a thin film of a 2D $Ti_{0.87}O_2$ nanosheet with sub-nm pores assembled on Celgard 2400, commercially available, monolayer PP membrane. The layers were self-assembled layer by layer using a vacuum filtration method. Multifunctional approaches to facilitate cationic (Li^+) and prevent anionic (polysulfide) migration in Li-S battery separators are well known using the principles of physisorption, chemisorption and electrostatic repulsion (Gupta and Sivaram, Energy Technol. 2019, 1800819). However, this paper is a step change in the strategy to combine selective sieving (pore-engineering) with electrostatic forces to improve the performance of the separator membrane. The properties reported for the composite membrane are attractive; no shrinkage at 120°C, improved surface wettability and reduced propensity to dendrite formation. The surface negative charge facilitates Li^+ transport while providing electrostatic repulsion for inhibiting polysulfide migration and the small pore size also allows only Li^+ to pass through (ion-sieving effect). The concept of tailoring a separator surface has been well described in the manuscript and the benefits that accrue to the operation of a Li-S batteries are supported by adequate data. The superiority of the approach is evident from the Table provided by the authors in the supplementary section. The cycling performance is superior to all known separators in the literature. This work will advance the field of tailored separator surfaces for demanding battery applications with special reference to Li-S batteries. Nevertheless, there are a few issues that the authors should address before the paper can be accepted for publication.

Response: We sincerely appreciate the reviewer for the positive comments on our work. In particular, we appreciate the reviewer's comments that "This work will advance the field of tailored separator surfaces for demanding battery applications with special reference to Li-S batteries". The mentioned reference has been cited in the revised manuscript.

Corresponding revision in References:

[25] Gupta, A. & Sivaram, S. Separator membranes for lithium–sulfur batteries: design principles, structure, and performance. *Energy Technol.* **7**, 1800819 (2019).

Comment 1: *The nature of adhesion between the polar $Ti_{0.87}O_2$ nanosheet and the PP surface has not been addressed. How stable are these surfaces for long term use?*

Response: Many thanks for reminding us to make it clear. The functional $\text{Ti}_{0.87}\text{O}_2/\text{PP}$ separators were prepared by vacuum filtration of a stable mixed suspension of $\text{Ti}_{0.87}\text{O}_2$ nanosheets and poly(tetrafluoroethylene) (PTFE) on a commercial PP separator (Clegard 2400). The PTFE was used as a binder to enhance the interfacial adhesion between nanosheets and PP surfaces. A strong adhesion between the $\text{Ti}_{0.87}\text{O}_2$ nanosheets and PP surfaces was possibly formed, which may be ascribed to the van der Waals interaction via the PTFE binder. As can be seen in Figure S18, the as-prepared $\text{Ti}_{0.87}\text{O}_2/\text{PP}$ separators showed a high stability under mechanical bending. Besides, after a long-term electrochemical cycling, the $\text{Ti}_{0.87}\text{O}_2$ nanosheets were still maintained on the PP separator (Figure S50). According to the reviewer's suggestions, the related discussion has been added in the revised manuscript.

Corresponding revision in Manuscript Page 13:

This suggested a strong adhesion between $\text{Ti}_{0.87}\text{O}_2$ nanosheets and PP separators, which may be ascribed to the van der Waals interaction produced by the poly(tetrafluoroethylene) (PTFE) binder.

Comment 2: *The process of deposition uses 5% PTFE and UV irradiation. The purpose of these treatment is not described.*

Response: Thanks for the reviewer's comment. The PTFE was used as a binder to enhance the interfacial adhesion between nanosheets and PP surfaces. The illumination with ultraviolet (UV) light in air aims to photocatalytically decompose the organic ions (such as TBA^+) surrounding the nanosheets, which has been widely used in our previous reports [*Chem. Mater.* **14**, 3524–3530 (2002); *Sci. Rep.* **3**, 2801 (2013); *J. Am. Chem. Soc.* **138**, 7621–7625 (2016); *J. Am. Chem. Soc.* **139**, 10868–10874 (2017); *ACS Nano* **12**, 12337–12346 (2018)]. According to the reviewer's suggestions, the related description has been added in the experimental section.

Corresponding revision in Experimental Section of Supplementary Information:

The PTFE was used as a binder to enhance the interfacial adhesion between nanosheets and PP surfaces.

The illumination with ultraviolet (UV) light in air aims to photocatalytically decompose the organic ions (such as TBA^+) surrounding the nanosheets.

Comment 3: *Lithium ion conductivity shows only a modest 25 % increase. Why?*

Response: We thank the reviewer’s comment. We summarized some previously reported works on modified separators for Li-S batteries. As shown in the Table 1 below, modification of pristine separators sometimes results in decrease of the Li⁺ conductivity because the coating layers may impede the Li⁺ diffusion pathway [*Adv. Mater.* **29**, 1606817 (2017); *Adv. Energy Mater.* **8**, 1802130 (2018); *Adv. Energy Mater.* **8**, 1802430 (2018)]. Here, the Li⁺ conductivity of the Ti_{0.87}O₂/PP separator is 0.381 mS cm⁻¹, which is higher than that of the pristine PP separator (0.305 mS cm⁻¹). Similar phenomenon of slightly increased conductivity for modified separators have also been observed in recent papers as shown in the table below [*Adv. Energy Mater.* **8**, 1801778 (2018); *Adv. Energy Mater.* **9**, 1901609 (2019)]. This is mainly due to the unique structure of the negatively charged Ti_{0.87}O₂ nanosheets with Ti vacancies. The electrostatic attraction force between Ti_{0.87}O₂ nanosheets and Li⁺-cations facilitates the migration of Li ions towards the membrane with subsequent diffusion through the membrane [*Adv. Mater.* **31**, 1900342 (2019); *Matter* **3**, 1685–1700 (2020)]. In addition, the nanolayered structure of the Ti_{0.87}O₂ nanosheets facilitates Li⁺ diffusion [*Adv. Energy Mater.* **8**, 1801778 (2018)]. The improved electrolyte wettability of the Ti_{0.87}O₂/PP separator also contributes to the higher Li⁺ conductivity [*Nano Energy* **59**, 390–398 (2019)]. More importantly, the Ti vacancies play a key role in boosting increased Li⁺ conductivity in our Ti_{0.87}O₂/PP separators by providing an expressway for rapid transportation of Li⁺ ions in addition to the conventional interlayer galleries between the nanosheets [*Science* **370**, 596–600 (2020); *Inorg. Chem.* **46**, 4787–4789 (2007); *Energy Environ. Sci.* **4**, 3509–3512 (2011)]. The related discussion has been added in the revised manuscript. The Table 1 below has been added in the supplementary information as Table S2.

Table 1. Comparison of Li⁺ conductivities of pristine and modified separators.

Modified separator	Li ⁺ conductivity mS cm ⁻¹	Pristine separator	Li ⁺ conductivity mS cm ⁻¹	Reference
MoS ₂ /Celgard	0.20	Celgard	0.33	Adv. Mater. 29 , 1606817 (2017)
LNS/CB-Celgard	0.590	Celgard	0.559	Adv. Energy Mater. 8 , 1801778 (2018)
MOF@PVDF - HFP	0.094	Celgard	0.138	Adv. Energy Mater. 8 , 1802130 (2018)
MoS ₂ -PDDA/PAA	0.48	Celgard	0.51	Adv. Energy Mater. 8 , 1802430 (2018)

Co-N _x @NPC/G - PP	0.684	PP	0.403	Adv. Energy Mater. 9 , 1901609 (2019)
Ti _{0.87} O ₂ /PP	0.381	PP	0.305	This work

Corresponding revision in Manuscript Page 13-14:

Table S2 summarized some previously reported work on modified separators for Li-S batteries. Generally, covering open pores of pristine separators will increase the pathway of ion movement, leading to reduced Li-ion diffusion. Thus, as shown in the Table S2, modification of pristine separators sometimes results in decrease of the Li⁺ conductivity. However, the above testing results demonstrated that Ti_{0.87}O₂ nanosheets can facilitate Li-ion migration. Similar phenomenon of slightly increased conductivity for modified separators have also been observed in recent papers as shown in the Table S2. Because the Ti_{0.87}O₂ layers are negatively charged with cation vacancies, the electrostatic attraction force between Ti_{0.87}O₂ nanosheets and Li⁺-cations facilitates the migration of Li ions towards the membrane with subsequent diffusion through the membrane.

Corresponding revisions in Supplementary Information:

Table S2. Comparison of Li⁺ conductivities of pristine and modified separators.

Modified separator	Li ⁺ conductivity mS cm ⁻¹	Pristine separator	Li ⁺ conductivity mS cm ⁻¹	Ref
MoS ₂ /Celgard	0.20	Celgard	0.33	30
LNS/CB-Celgard	0.590	Celgard	0.559	50
MOF@PVDF - HFP	0.094	Celgard	0.138	47
MoS ₂ -PDDA/PAA	0.48	Celgard	0.51	31
Co-N _x @NPC/G - PP	0.684	PP	0.403	22
Ti _{0.87} O ₂ /PP	0.381	PP	0.305	This work

Comment 4: *Celgard 2400 is known to possess wide slit-like macropores. Obviously the nanosheet will penetrate these pores and block the wide pores of the PP. This is evident from an examination of Figures S3 and S6. What would be the consequence of this to the flux of the electrolyte?*

Response: We appreciate the reviewer's comments. As shown in Figure S3, the pristine PP separator possess wide slit-like macropores in a pore size of 50–100 nm. As shown in Figure 2b, the $\text{Ti}_{0.87}\text{O}_2$ nanosheets have an average lateral size of 500 nm, which is larger than the size of the pores on PP separator. Thus, the $\text{Ti}_{0.87}\text{O}_2$ nanosheets uniformly cover the surfaces of the PP separator, as shown in Figure 2h and Figure S8 in the revised manuscript. Although the nanosheets covered the wide pores of PP separators, it seems unlikely that these covered nanosheets on PP separators would block the flux of electrolytes. Figure S17 shows the contact angles of electrolyte on the PP and $\text{Ti}_{0.87}\text{O}_2/\text{PP}$ separators. A smaller contact angle was observed on the $\text{Ti}_{0.87}\text{O}_2/\text{PP}$ separators than that on PP separators, suggesting a better wettability of the $\text{Ti}_{0.87}\text{O}_2/\text{PP}$ separators toward electrolyte. Besides, curved nanowrinkles without obvious cracks were observed for the obtained $\text{Ti}_{0.87}\text{O}_2$ layers on PP separators (Figure 2h), which often results in additional and wider transporting channels for increased permeability [*ACS Nano* **14**, 2137–2144 (2020); *Nat. Commun.* **4**, 2979 (2013)]. These results suggest that the $\text{Ti}_{0.87}\text{O}_2$ layer is beneficial for accelerating the electrolyte penetration, thus facilitating the transport of Li^+ ions. Similar results have been reported in other modified separators, which were covered by different functional nanostructures [*ChemSusChem* **11**, 3345–3351 (2018); *J. Mater. Chem. A* **6**, 7375–7381 (2018); *Nanoscale* **10**, 13694–13701 (2018); *Adv. Energy Mater.* **9**, 1901609 (2019)]. The related discussion has been added in the revised manuscript.

Corresponding revisions in Manuscript Page 9:

Curved nanowrinkles without obvious cracks were observed for the obtained $\text{Ti}_{0.87}\text{O}_2$ layers on PP separators, which often results in additional and wider transporting channels for increased permeability^{67,68}.

Corresponding revisions in Manuscript Page 13:

Although the nanosheets covered the wide pores of PP separators, the improved wettability is beneficial for accelerating the electrolyte penetration, thus facilitating the transport of Li^+ ions²¹.

Corresponding revisions in References:

[21] Cheng, Z. *et al.* Separator modified by cobalt-embedded carbon nanosheets enabling chemisorption and catalytic effects of polysulfides for high-energy-density lithium-sulfur batteries. *Adv. Energy Mater.* **9**, 1901609 (2019).

[67] Zheng, F. *et al.* Critical stable length in wrinkles of two-dimensional materials. *ACS Nano* **14**, 2137–2144 (2020).

[68] Huang, H. *et al.* Ultrafast viscous water flow through nanostrand-channelled graphene oxide membranes. *Nat. Commun.* **4**, 2979 (2013).

Comment 5: *Celgard 2400 do not possess tortuous pores. Therefore, covering the open pores of Celgard 2400 will not increase the tortuosity as stated in p.12.*

Response: Thanks for the reviewer's comment. We agree with the reviewer that it is inappropriate to say that covering the open pores of PP separators will increase the tortuosity of ion movement. This discussion in original manuscript has been deleted. Indeed, after covering of the pores of PP separators with nanosheets, the ions have to diffuse through the tortuous interlayers between the nanosheets before moving through the pores of PP separators. Thus, the pathway of ion movement was increased. The description had been revised to "covering open pores of pristine separators will increase the pathway of ion movement".

Corresponding revision in Manuscript Page 13:

Generally, covering open pores of pristine separators will increase the pathway of ion movement, leading to reduced Li-ion diffusion.

Comment 6: *Maximum lithium ion conductivity is observed at 0.016 mg/cm² weight density. Therefore, it is necessary to show the SEM of this sample, which is missing.*

Response: Thanks for the reviewer's comment. Actually, in the Experimental section of Supplementary Information, we had mentioned that the weight density of the Ti_{0.87}O₂ nanosheets in the Ti_{0.87}O₂/PP separator was estimated to be ~0.016 mg cm⁻². The Figures 2h and 2i are the SEM images of the Ti_{0.87}O₂/PP separator with a weight density of 0.016 mg cm⁻². Following the reviewer's suggestion, we have added the above description in the revised manuscript.

Corresponding revision in Manuscript Page 10:

The weight density of the Ti_{0.87}O₂ nanosheets in the Ti_{0.87}O₂/PP separator was estimated to be ~0.016 mg cm⁻².

Reviewer #3:

The manuscript entitled “Atomic-scale tandem regulation of anionic and cationic migration for long-life alkali metal batteries” deals with modification of the PP separator with negatively charged Ti_xO_2 nanosheets with cation vacancies on the cathodic and anodic side of Li-S and Na-Se batteries in order to support the diffusion of smaller ions like Li^+/Na^+ or S^{2-}/Se^{2-} , but to impede the diffusion of polysulfides and polyselenides. The manuscript implies a lot of characterization methods and is very comprehensive. However, the results obtained in present work are comparable with other results on the field of Li-S batteries in terms of specific capacity, long-term cycling and a sulfur loading. Thus, the average value of about 650 mAh g⁻¹ at the current density of 1C was obtained for 5000 cycles, while about 350 mAh g⁻¹ at 2C for the next 10000 cycles, with the mass loading of 3.5 mg cm⁻². According to the literature, comparable characteristics (high mass loading and stable cycling behavior) are already known for Li-S batteries. For example, G. Zhou et al. reported in 2015 about specific capacity of 500 mAh g⁻¹ at 0.9C after 1000 cycles with a much higher sulfur mass loading of 10.1 mg cm⁻² (Nano Energy 2015, 11, 356) using the graphene foam S-cathode. The graphene foam provided an electrically conductive network, robust mechanical support and sufficient space for a high sulfur loading. Next, a protective coating of the PP separator by mesoporous carbon layer on the cathodic side allowed localization of dissolved polysulfide intermediates and retained them as active material within the cathode side, suppressing their further diffusion to the anodic side (Adv. Funct. Mater. 2015, 25, 5285). A stable cycling with specific capacity of 900 mAh g⁻¹ at 1C for 500 cycles was observed. Further, a 2D protective MoS₂ anodic layer, coated on the Li-anode, enables specific capacity of about 1000 mAh g⁻¹ at 0.5C for at least 1200 cycles (Nature Nanotechnology, 2018, 13, 337). A uniform Li-deposition and stripping without dendrite formation and suppression of polysulfides diffusion through the 2D layers to the anode side were the reasons for the observed stable cycling behavior. Therefore, the results from the current manuscript represent a further improvement of the already existing strategies for enhancement of the Li-S battery performance and are not novel. I believe that due to this reason the manuscript is not suitable for Nat. Comm. and must be transferred to a more technological journal.

Response: Thanks for your efforts to review our manuscript in detail. We appreciate the reviewer’s comments that “the manuscript implies a lot of characterization methods and is very comprehensive”. We understand that the reviewer may concern about the novelty of our work. The reviewer had mentioned three references and concluded that the electrochemical performances obtained in our work are comparable with other results in terms of specific capacity, long-term

cycling and a sulfur loading. Although we agree with the reviewer that these references reported high electrochemical performances, it should be noted that the main innovation of our work is not to obtain high specific capacities at a high sulfur mass loading. We thank the reviewer for reminding us to explain this issue clearly. A response is provided to distinguish our work from the previously reported works as detailed below.

(1) Our work reported a multifunctional approach that can simultaneously relieve the issues of both S cathodes and Li anodes for Li-S batteries

Li-S batteries are considered as one of the most promising candidates for next-generation energy storage due to their high theoretical energy density. However, multiple obstacles associated with both S cathodes and Li anodes have severely hindered their practical applications, especially, the shuttling of soluble polysulfide (PS) intermediates and the formation of Li dendrites.

To tackle these detrimental issues, various strategies have been developed to enhance the electrochemical properties of Li-S batteries by either impeding the shuttling effect or suppressing dendrite growth. For example, in the reference [*Nano Energy* **11**, 356–365 (2015)] that the reviewer cited, a graphene foam was used as a sulfur host to prepare a composite S cathode. The highly conductive network, robust mechanical properties and sufficient space of the graphene foam ensure a high sulfur loading and a high areal capacity. In the reference [*Adv. Funct. Mater.* **25**, 5285–5291 (2015)] that the reviewer cited, a mesoporous carbon layer was used to modify the separator for suppressing the migration of soluble PSs, which thus enhances the overall electrochemical performance of Li-S batteries. However, in these two reports [*Nano Energy* **11**, 356–365 (2015); *Adv. Funct. Mater.* **25**, 5285–5291 (2015)] that the reviewer cited, the issues associated with Li anodes still remain a major concern. Besides, in the reference [*Nat. Nanotechnol.* **13**, 337–344 (2018)] that the reviewer cited, atomic layers of two-dimensional (2D) MoS₂ were used to coated on Li metal. The MoS₂ layer exhibits tight adhesion to the surface of Li metal and facilitates uniform flow of Li⁺ into and out of bulk Li metal. Thus, a stable Li anode is realized with effectively suppressed formations of Li dendrite. However, this modification strategy of Li metal with 2D MoS₂ [*Nat. Nanotechnol.* **13**, 337–344 (2018)] that the reviewer cited could not solve the problem of S cathodes. A three-dimensional (3D) carbon nanotube (CNT) composite structures were needed to use as a sulfur host to prepare a CNT-S cathode in the above report that the reviewer pointed out, which improve the electrical conductivity of the sulfur cathode and prevent PSs from shuttling during cycling.

Here, in this work, we aim to develop a multifunctional approach which can simultaneously improve both S cathodes and Li anodes. On the one hand, the unwanted shuttle effect of soluble PSs and the formation of Li dendrites are originated from the uneven migration of PS anions and Li cations [*Acc. Chem. Res.* **46**, 1125–1134 (2013); *Acc. Chem. Res.* **46**, 1135–1143 (2013); *Chem. Soc. Rev.* **45**, 5605–5634 (2016)]. On the other hand, in a battery system, separator membranes provide channels for diffusion of both cations and anions. Inspired by the above in-depth understanding, we propose that modification of separators could be a straightforward way to control the migration of both PS anions and Li cations, which can simultaneously eliminate the shuttle effect of PSs and the formation of Li dendrites.

(2) Our work reported an optimized $\text{Ti}_{0.87}\text{O}_2$ layer with an ultralow weight density and an ultrathin thickness, which has not been reported previously

So far, various functional materials have been employed to modify separators, including carbon materials, metal-based oxides, sulfides, carbides and hydroxides, and metal-organic frameworks (MOFs). However, to effectively suppress PS shuttling, most of the functional layers used to date have a high weight density and a large thickness. For example, in the reference [*Adv. Funct. Mater.* **25**, 5285–5291 (2015)] that the reviewer cited, a mesoporous carbon layer with a thickness of ~ 27 μm was used to modify the separator, as shown in the Figure 1 below. The mass loading of the coated mesoporous carbon layer was 0.5 mg cm^{-2} , which is half weight of the pristine separator, as shown in the Table 2 below. Several other 2D atomically-thin nanosheets have also been employed to modify the surfaces of separators. However, the functional layers of these separators generally have a thickness ranging from several micrometers to hundreds of micrometers and a large mass loading, as shown in the Table 2 below. For all above reports, although the migration of soluble PSs was suppressed, the thick functional layer was an extra barrier to ion transfer, which causes large interfacial resistance and suppresses the transport of Li metal cations. Besides, these inevitably bring a severe burden in terms of the weight and useable volume of the whole cell, which subtracts from the targeted high energy densities of Li-S batteries. In our work, an optimized $\text{Ti}_{0.87}\text{O}_2$ layer shows an ultralow weight density of 0.016 mg cm^{-2} and an ultrathin thickness of $\sim 80 \text{ nm}$. It should also be noted that the weight and thickness of the $\text{Ti}_{0.87}\text{O}_2$ layer was only approximately 0.32% and 1.5% of those of the commercial PP separator (thickness, $25 \mu\text{m}$; weight, 2.16 mg ; diameter, 16 mm), respectively. This ultrathin $\text{Ti}_{0.87}\text{O}_2$ layer maximizes Li-metal cation transport without compromising the ability to prevent PS shuttling. To the best of our knowledge, such a low weight density and

ultrathin thickness have not been reported previously (Table S1 in the revised Supplementary information).

Table 2. Comparison of weight density and thickness of functional layers on separators.

Functional materials	Weight density mg cm ⁻²	Thickness μ m	Reference
Mesoporous carbon	0.5	27	Adv. Funct. Mater. 25 , 5285–5291 (2015) (the reference that the reviewer cited)
Graphene oxide	0.12	5	ACS Nano 9 , 3002–3011 (2015)
Graphene	1.3	30	Adv. Mater. 27 , 641–647 (2015)
MoS ₂ -PDDA/PAA	0.1	3	Adv. Energy Mater. 8 , 1802430 (2018)
Ti ₃ C ₂ MXene	0.1	0.522	ACS Appl. Mater. Interfaces 8 , 29427–29433 (2016)
Black Phosphorus	0.4	~0.35	Adv. Mater. 28 , 9797–9803 (2016)
Cu ₂ (CuTCPP) nanosheets	0.1	0.5	Energy Storage Mater. 21 , 14–21 (2019)
Laponite nanosheets	0.7	3.5	Adv. Energy Mater. 8 , 1801778 (2018)
Ti _{0.87} O ₂	0.016	0.080	This work

Figure 1. Left: in the reference [Adv. Funct. Mater. 25, 5285–5291 (2015)] that the reviewer cited, a cross-sectional SEM image shows that a mesoporous carbon layer with a thickness of $\sim 27 \mu\text{m}$ was used to modify the separator. Scale bar: $20 \mu\text{m}$. **Right:** in our work, a cross-sectional SEM image shows that the $\text{Ti}_{0.87}\text{O}_2$ layer has a ultrathin thickness of $\sim 80 \text{ nm}$.

(3) For the first time, the unilamellar nanosheets with atomic vacancies were used for tandem regulation of anionic and cationic migration

An ideal functional layer for Li-S batteries should be as thin as possible to maximize Li-metal cation transport without compromising the ability to prevent PS shuttling, thus forming a selective ionic sieve with high ion permeability. In this regard, 2D atomically-thin nanosheets with sub-nanometer pores are attractive as a highly selective and permeable separator for long-life Li-S batteries. Several 2D atomically-thin nanosheet materials had been investigated, including graphene oxide, MoS_2 and MXenes. The PS shuttling effects were mitigated owing to the steric hindrance effect, but the diffusion of Li ions was also hindered. Furthermore, to certain extent, without nanopores, the Li^+ ions could only migrate through interlayers between the nanosheets.

Here, for the first time, 2D unilamellar $\text{Ti}_{0.87}\text{O}_2$ nanosheets with atomic Ti vacancies were reported as a selective ionic sieve for attaining high-performance Li-S, Li-Se and Na-Se batteries. The $\text{Ti}_{0.87}\text{O}_2$ nanosheet is a single-crystal-like 2D monolayer (0.75 nm thickness) with a high density of Ti vacancies. The Ti atomic vacancy is the octahedral site, which can accommodate a Li^+ ion (0.76 Å diameter) or Na^+ ion (1.02 Å diameter), but smaller than a PS anion. Therefore, the Ti atomic vacancies can allow small Li^+/Na^+ ions to rapidly pass through, which efficiently exclude large PS anions. Besides, the Ti cation vacancies endow the obtained nanosheets with negative charges. The negatively charged $\text{Ti}_{0.87}\text{O}_2$ nanosheets can effectively exclude PS anions via a strong electrostatic repulsion effect, therefore simultaneously offering strong electrostatic interaction for the

efficient adhesion and homogeneous distribution of Li^+/Na^+ ion flux, resulting in effective elimination of Li/Na dendrite growth.

(4) Our work reported the best cycling stability among reported functionalized separators for Li-S batteries

In order to highlight the function of $\text{Ti}_{0.87}\text{O}_2$ nanosheets, we used carbon black as the cathode matrix, which has almost no PS adsorption ability. The assembled Li-S cells with the $\text{Ti}_{0.87}\text{O}_2/\text{PP}$ separators delivered a long-term cycle stability with an ultralow capacity decay of 0.0036% per cycle for over 5000 cycles (Figure 6b). Even with an increased sulfur loading, the cell was still able to deliver an ultralow capacity decay of 0.0035% per cycle for 5000 cycles at 1C and an ultralow capacity decay of 0.0035% per cycle for over 10000 cycles at 2C (Figure 6d).

Various 2D nanosheets have been employed to fabricate the functionalized separators for Li-S batteries. After a comprehensive comparison, our work showed the best cycling stability among these reported functionalized separators for Li-S batteries (Figure 6c and Table S1), including graphene, layered double hydroxides (LDHs), MoS_2 , MXenes, and MOF, etc [Ref. 19, 23, 31, 32, 34–36, 40, 42–44, 46–49 in the revised manuscript]. Furthermore, the $\text{Ti}_{0.87}\text{O}_2/\text{PP}$ separators are also promising to improve the cycling stability for Li-Se and Na-Se batteries.

On the basis of the above discussion, we believe that **our work is the first report on a new strategy of using 2D unilamellar nanosheets with atomic vacancies to achieve tandem control of migration of both cations and anions in advanced batteries such as Li-S, Li-Se and Na-Se batteries.** As the second reviewer pointed out, “This work will advance the field of tailored separator surfaces for demanding battery applications with special reference to Li-S batteries”. The reference [*Nat. Nanotechnol.* **13**, 337–344 (2018)] that the reviewer pointed out had already been cited in our original manuscript. The other two references [*Nano Energy* **11**, 356–365 (2015); *Adv. Funct. Mater.* **25**, 5285–5291 (2015)] that the reviewer mentioned have been cited in the revised manuscript.

Corresponding revisions in References:

[14] Zhou, G. et al. A graphene foam electrode with high sulfur loading for flexible and high energy Li-S batteries. *Nano Energy* **11**, 356–365 (2015).

[26] Balach, J. et al. Functional mesoporous carbon-coated separator for long-life, high-energy lithium-sulfur batteries. *Adv. Funct. Mater.* **25**, 5285–5291 (2015).

Comment 1: *In the introduction part, the authors write “At the S/Se cathode side of alkali metal batteries (Figure 1d), the negatively charged $Ti_{0.87}O_2$ nanosheets with a high negative charge density effectively exclude PS anions via a strong electrostatic repulsion effect.” It would be more correct to write that PS anions cannot go through the $Ti_{0.87}O$ nanosheets primary because of the geometrical restrictions. A comparison of the anionic size for sulfide anion $S(2-)$ and for example for polysulfide anion $S_4(2-)$ (approximately 1.84 Å vs. 3.7-4.0 Å) speaks rather for impossible diffusion of polysulfide anions through the defect sites of $Ti_{0.87}O$ nanosheets. Moreover, the local negative charge density is higher for $S(2-)$ than for polysulfides, since in PS the charge is distributed within the bigger molecule. Therefore, the electrostatic repulsion must be higher in case of $Ti_{0.87}O$ nanosheets and $S(2-)$.*

Response: Many thanks for the reviewer’s valuable comments. We absolutely agree with the reviewer’s comments that the PS anions cannot go through the $Ti_{0.87}O_2$ nanosheets because of the geometrical restrictions. As we had already stated in the manuscript, the Ti atomic vacancy can accommodate a Li^+ ion (0.76 Å diameter) or Na^+ ion (1.02 Å diameter) but exclude a PS anion. Therefore, the Ti vacancies function as migration-aids for Li^+/Na^+ ions and obstacle channels for PS anions, respectively. According to the reviewer’s suggestion, the related discussion has been added in the introduction part.

Corresponding revisions in Manuscript Page 7:

Besides, the PS anions with sizes larger than the size of the Ti vacancies are selectively excluded because of the geometrical restrictions.

Comment 2: *Did authors confirm the composition of $Ti_{0.8}O$ nanosheets by the chemical analysis? An XRD analysis of $Ti_{0.8}O$ nanosheets without PP separator would be useful as well. What about hydrogen content in the nanosheets? According to the stoichiometry of $Ti_{0.8}O$, there is a quite large negative charge on the surface. Could authors exclude the formation of $TiO_2 \cdot xH_2O$ with structural water, which cannot be removed during drying process under soft conditions?*

Response: We appreciate the reviewer’s valuable suggestion. The $Ti_{0.87}O_2$ nanosheets were prepared based on a soft chemical exfoliation process of layered lepidocrocite-type titanate crystals, which has been reported by us [*Chem. Mater.* **7**, 1001–1007 (1995); *J. Am. Chem. Soc.* **118**, 8329–8335 (1996); *Chem. Mater.* **9**, 602–608 (1997); *J. Am. Chem. Soc.* **120**, 4682–4689 (1998)]. The layered lepidocrocite-type titanate crystals, with a chemical composition of $K_{0.8}Ti_{1.73}Li_{0.27}O_4$, were obtained

by a solid-phase reaction. Then, after an acid-exchange treatment, protonated layered titanate crystals ($\text{H}_{1.07}\text{Ti}_{1.73}\text{O}_4 \cdot \text{H}_2\text{O}$) were collected. Subsequently, a delamination process was conducted by shaking the protonated crystal in a tetrabutylammonium (TBA^+) hydroxide aqueous solution. Because the delamination was carried out at mild conditions, it is reasonable to assume that the composition does not change. Thus, the composition of delaminated nanosheet in a suspension is $\text{Ti}_{0.87}\text{O}_2$ ($\text{Ti}_{0.87}\square_{0.13}\text{O}_2$, where \square represents the Ti vacancies) [*Chem. Mater.* **10**, 4123–4128 (1998); *Nat. Commun.* **4**, 1632 (2013); *J. Am. Chem. Soc.* **136**, 5491–5500 (2014)]. The presence of Ti vacancies in the nanosheets was confirmed by the high-angle annular dark-field STEM (HAADF-STEM) observation [*Sci. Rep.* **3**, 2801 (2013); *ACS Nano* **12**, 12337–12346 (2018)]. Generally quantitative analysis of O is not done for oxides because of difficulty. Many characterizations are compatible with this nanosheet composition, although direct quantization of Ti/O was not done [*J. Am. Chem. Soc.* **118**, 8329–8335 (1996); *J. Am. Chem. Soc.* **120**, 4682–4689 (1998); *Nat. Commun.* **4**, 1632 (2013); *J. Am. Chem. Soc.* **136**, 5491–5500 (2014); *Chem. Rev.* **114**, 9455–9486 (2014)]. Thus, it seems unlikely to form the $\text{TiO}_2 \cdot x\text{H}_2\text{O}$ that the reviewer suspects. Besides, the nanosheets are negatively charged as confirmed by zeta-potential measurements (Figure S1), which is another evidence excluding the formation of electrically neutral $\text{TiO}_2 \cdot x\text{H}_2\text{O}$.

According to the reviewer's suggestion, an XRD analysis of $\text{Ti}_{0.87}\text{O}_2$ nanosheets without PP separator was conducted. As shown in Figure S5, two diffraction peaks appeared in a low angular range can be indexed as 010 and 020. This indicates a lamellar structure with a gallery height of ~ 1.1 nm, which is consistent with the results of self-assembled $\text{Ti}_{0.87}\text{O}_2$ nanosheets on PP separators. All the related discussions have been added in the revised manuscript.

Besides, we agree with the reviewer that, due to the negative charge on the surface, some positively charged species should be included in the $\text{Ti}_{0.87}\text{O}_2$ nanosheets to make the whole charge balance. We had reported the intercalation/delamination behaviour of lepidocrocite-type titanate in composition of $\text{H}_{0.8}\text{Ti}_{1.2}\text{Fe}_{0.8}\text{O}_4 \cdot \text{H}_2\text{O}$ with various amines and ammonium ions, such as TBA^+ ions [*J. Am. Chem. Soc.* **136**, 5491–5500 (2014)]. About 30% of exchangeable protons was replaced with TBA^+ ions during the swelling-exfoliation process. Thus, after isolating from the suspension, the nanosheets have a composition of $\text{H}_{0.56}\text{TBA}_{0.24}\text{Ti}_{1.2}\text{Fe}_{0.8}\text{O}_4 \cdot x\text{H}_2\text{O}$. The reactivity of layered titanate $\text{H}_{1.07}\text{Ti}_{1.73}\text{O}_4 \cdot \text{H}_2\text{O}$ used in our work was similar to that of $\text{H}_{0.8}\text{Ti}_{1.2}\text{Fe}_{0.8}\text{O}_4 \cdot \text{H}_2\text{O}$. Thus, after filtering on PP separator, the nanosheet films should have a composition of $\text{H}_{0.75}\text{TBA}_{0.32}\text{Ti}_{1.73}\text{O}_4 \cdot x\text{H}_2\text{O}$. After illumination under ultraviolet (UV) light, the small organic ions (such as TBA^+) surrounding the nanosheets were decomposed, which has been widely demonstrated in our previous reports [*Chem.*

Mater. **14**, 3524–3530 (2002); *Sci. Rep.* **3**, 2801 (2013); *J. Am. Chem. Soc.* **138**, 7621–7625 (2016)]. In order to further analyse the composition, the weight loss of the nanosheets without PP separators was recorded during a heating process in air. As shown in Figure S6, the weight loss before 250 °C should be the liberation of structural H₂O molecules, while the decomposition of interlayer TBA⁺ ions should be responsible for the mass loss between 250 and 500 °C. Then, after a rough calculation, the final composition of the nanosheet films in our work was qualitatively determined to be H_{0.98}TBA_{0.09}Ti_{1.73}O₄·1.58H₂O. Considering that the interlayer distance of 1.1 nm is too small to accommodate TBA⁺ ions, the small amount of TBA⁺ ions should be included in a gap between the restacked Ti_{0.87}O₂ layers.

Corresponding revisions in Manuscript Page 9:

An XRD analysis of Ti_{0.87}O₂ nanosheets without PP separator was also conducted. As shown in Figure S5, two diffraction peaks appeared in a low angular range can be indexed as 010 and 020. This indicates a lamellar structure with a gallery height of ~1.1 nm, which is consistent with the results of self-assembled Ti_{0.87}O₂ nanosheets on PP separators.

Corresponding revisions in Manuscript Page 9-10:

Due to the negative charge on the surface, some positively charged species could be included in the Ti_{0.87}O₂ nanosheets to make the whole charge balance, such as proton and tetrabutylammonium (TBA⁺) ions⁶⁹. Almost all TBA⁺ ions initially trapped between the nanosheets could be decomposed upon exposure to UV light. In order to analyse the composition, the weight loss of the final nanosheet films without PP separators was recorded during a heating process in air. As shown in Figure S6, the weight loss before 250 °C is associated with the liberation of structural H₂O molecules; while the decomposition of interlayer TBA⁺ ions could result in the mass loss between 250 and 500 °C. Then, after a rough calculation, the final composition of the nanosheet films in our work was qualitatively determined to be H_{0.98}TBA_{0.09}Ti_{1.73}O₄·1.58H₂O. Considering that the interlayer distance of 1.1 nm is too small to accommodate TBA⁺ ions, the small amount of TBA⁺ ions should be included in a gap between the restacked Ti_{0.87}O₂ layers.

Corresponding revisions in Supplementary Information:

Figure S5. XRD pattern for the $\text{Ti}_{0.87}\text{O}_2$ nanosheets without PP separators.

Figure S6. Thermogravimetric curve for the nanosheet films without PP separators.

Corresponding revisions in References:

[69] Geng, F. et al. Gigantic swelling of inorganic layered materials: a bridge to molecularly thin two-dimensional nanosheets. *J. Am. Chem. Soc.* **136**, 5491–5500 (2014).

REVIEWERS' COMMENTS

Reviewer #2 (Remarks to the Author):

I have carefully gone through the response of the Author to my comments and the necessary modifications made in the manuscript.

I am satisfied with the response

The manuscript may be accepted for publication

S.Sivaram

Reviewer #3 (Remarks to the Author):

I carefully analyzed the response of the authors to my questions/remarks and came to the conclusion that the manuscript can now be published in Nature Communications.

Response to Reviewers' Comments

We would like to thank all reviewers for taking time and efforts to review our manuscript. We sincerely appreciate all the reviewers for their valuable comments and suggestions, which helped us to improve the overall quality of the manuscript.

Reviewer #2:

I have carefully gone through the response of the Author to my comments and the necessary modifications made in the manuscript. I am satisfied with the response. The manuscript may be accepted for publication

Response: We thank the reviewer for a constructive review process as well as strong support on the publication of this work.

Reviewer #3:

I carefully analyzed the response of the authors to my questions/remarks and came to the conclusion that the manuscript can now be published in Nature Communications.

Response: We thank the reviewer for a constructive review process as well as strong support on the publication of this work.